# Boosting Adversarial Robustness with CLAT: Criticality-Leveraged Adversarial Training

## Abstract

Adversarial training (AT) is a common technique for enhancing neural network robustness. Typically, AT updates all trainable parameters, but such comprehensive adjustments can lead to overfitting and increased generalization errors on clean data. Research suggests that fine-tuning specific parameters may be more effective; however, methods for identifying these essential parameters and establishing effective optimization objectives remain unclear and inadequately addressed. We present CLAT, an innovative adversarial fine-tuning algorithm that mitigates adversarial overfitting by integrating "criticality" into the training process. Instead of tuning the entire model, CLAT identifies and fine-tunes fewer parameters in robustness-critical layers—those predominantly learning non-robust features—while keeping the rest of the model fixed. Additionally, CLAT employs a dynamic layer selection process that adapts to changes in layer criticality during training. Empirical results demonstrate that CLAT can be seamlessly integrated with existing adversarial training methods, enhancing clean accuracy and adversarial robustness by over 2% compared to baseline approaches.

## 1 Introduction

Advancements in deep learning models have markedly improved image classification accuracy. Despite this, their vulnerability to adversarial attacks — subtle modifications to input images that mislead the model — remains a significant concern (Goodfellow et al., 2015; Szegedy et al., 2014). The research community has been rigorously exploring theories to comprehend the mechanics behind adversarial attacks (Bai et al., 2021). Ilyas et al. (2019) uncover the coexistence of robust and non-robust features in standard datasets. Adversarial vulnerability largely stems from the presence of non-robust features in models trained on standard datasets, which, while highly predictive and beneficial for clean accuracy, are susceptible to noise (Szegedy et al., 2014). Unfortunately, it is observed that deep learning models tend to preferentially learn these non-robust features. Inkawhich et al. (2019; 2020) further demonstrate that adversarial images derived from the hidden features of certain intermediate non-robust/"critical" layers exhibit enhanced transferability to unseen models. This suggests a commonality in the non-robust features captured by these layers. While identifying these critical layers to improve their robustness is appealing, this process often requires the time-consuming generation of attacks against each individual layer. Methods to identify and effectively address the criticality of such layers are still lacking.

In contrast to layer-wise feature vulnerability analysis, adversarial training (Athalye et al., 2018; Madry et al., 2019; Croce & Hein, 2020), involves training entire neural networks with adversarial examples generated in real-time. This approach inherently encourages all layers in the model to learn

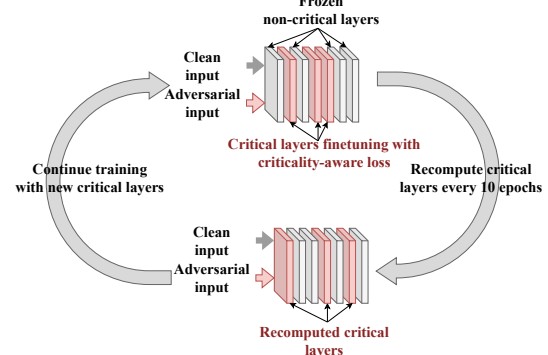

Figure 1: **CLAT overview.** CLAT fine-tunes the selected critical layers (red) while freezing other layers (grey). fine-tuning objective is computed per Eq. (6). Critical layers are adjusted periodically. Pseudocode is provided in **Appendix A**.

robust features from adversarial images, thereby enhancing the model's resilience against attacks. However, given the more challenging optimization process of learning from adversarial examples than from clean ones, adversarial training also brings hurdles such as heightened errors on clean data and susceptibility to overfitting, ultimately diminishing its effectiveness in practical applications (Schmidt et al., 2018; Zhang et al., 2019; Raghunathan et al., 2019; Javanmard et al., 2020). Despite various efforts to enhance adversarial training, such as modifying input data and adjusting loss functions (Hitaj et al., 2021; Raghunathan et al., 2019; Zhang et al., 2019; Wang et al., 2020; Wu et al., 2020b; Pang et al., 2022), these approaches still frequently fall short in alleviating the aforementioned issues.

In light of these challenges, we introduce CLAT, a paradigm shift in adversarial training, where we mitigate overfitting during adversarial training by identifying and tuning only the robustness-critical model layers. CLAT commences by pinpointing critical layers within a model using our novel, theoretically grounded, and easily computable,**"criticality index"**, which we developed to identify layers which have learned non-robust features dominantly. Subsequently, our algorithm meticulously fine-tunes these critical layers to remove their non-robust features and reduce their criticality, while freezing the other, non-critical layers. Dynamic selection of critical layers is conducted during the training process to always focus fine-tuning on the most-in-need layers, avoiding the overfitting of full-model adversarial training. CLAT therefore achieves both clean and adversarial state-of-the-art (SOTA) accuracy compared to previous adversarial training methods.

In summary, we make the following contributions in this paper:

- We introduce the "criticality index", a quantitative metric designed to identify critical layers for the adversarial vulnerability of a model with minimal overhead.
- We develop a specialized adversarial training objective focused on reducing the criticality of the identified critical layers to bolster overall model robustness.
- We propose CLAT, an adversarial fine-tuning algorithm that mitigates overfitting by focusing on reducing the criticality of fewer than 4% of trainable parameters. CLAT integrates seamlessly into diverse model training scenarios and baseline adversarial training methods.

CLAT stands out by markedly reducing overfitting risks, boosting both clean accuracy and adversarial resilience by approximately *2%*.

## 2 RELATED WORK

### 2.1 ADVERSARIAL TRAINING

Adversarial Training (AT) was first introduced by (Goodfellow et al., 2015), who demonstrated how the integration of adversarial examples into the training process could substantially improve model robustness. This idea evolved into a sophisticated minimax optimization approach with Projected Gradient Descent Adversarial Training (PGD-AT) (Madry et al., 2019), which employs PGD attacks in training. Regarded as the gold standard in AT, PGD-AT generates adversarial training samples using multiple steps of projected gradient descent, leading to substantially improved empirical robustness (Carlini & Wagner, 2017; Athalye et al., 2018; Croce & Hein, 2020). Further refining this approach, TRADES (Zhang et al., 2019) optimizes a novel loss function to balance classification accuracy with adversarial robustness. Recent enhancements in AT, including model ensemble and data augmentation, have also produced notable improvements in model resilience. (Xie et al., 2020; Yang et al., 2020; Carmon et al., 2022). Inkawhich et al. (2019) propose "Activation Attacks" (AA) which underscore the efficacy of leveraging intermediate model layers for generating stronger adversarial attacks, suggesting that incorporating AA in adversarial training could fortify defenses. Their findings provide a foundation for our method which integrates these intermediate critical layers into our adversarial training strategy.

### 2.2 ADVERSARIAL TRAINING IMPROVEMENTS AND ROBUST OVERFITTING

Adversarial training methods like PGD-AT and TRADES are computationally expensive and prone to overfitting, requiring multi-step adversary generation, complex objectives, and extensive model tuning (Shafahi et al., 2019). To improve efficiency, Shafahi et al. (2019) proposed "Free" AT, which accelerates training by using a single backpropagation step for both training and PGD adversary

generation. However, gradient alignment issues led to reduced robustness and increased overfitting. Similarly, Wong et al. (2020) introduced "Fast" AT, but it also suffered from similar weaknesses (Andriushchenko & Flammarion, 2020), prompting the development of GradAlign. Unfortunately, GradAlign tripled training time due to second-order gradient computation. Later efforts sought to address robust overfitting. RiFT (Zhu et al., 2023) improved general performance by leveraging layer redundancies but was constrained by heuristic redundancy measurements. Zhang et al. (2024) mitigated overfitting by disentangling natural and adversarial objectives, yet model-wide adjustments still limited robustness. In contrast, CLAT uses a theoretically grounded, dynamic, critical layer selection mechanism, resulting in improved robust generalization by tuning a critical subset of layers. Furthermore, CLAT is agnostic to attack generation methods in the AT processing, making it an ideal complement to existing fast-AT methods to mitigate overfitting.

## 3 METHODS

Building on prior attack and defense research (Inkawhich et al., 2019; 2020; Zhu et al., 2023) which demonstrates that not all model layers equally learn non-robust features and having all layers learn robust features leads to overfitting, we aim to improve model robustness by identifying and fine-tuning only those critical layers that are prone to learning non-robust features, while keeping the non-critical layers frozen. In this section, we begin by defining and identifying critical layers, then outline our objectives for reducing their criticality. Finally, we present our complete CLAT algorithm, which effectively mitigates overfitting and enhances model robustness.

### 3.1 LAYER CRITICALITY

Consider a deep learning model with $n$ layers, and an input $x$, defined as:

$$F(x) = f_n(f_{n-1}(\ldots f_1(x))), \tag{1}$$

where the functionality of the $i$-th layer is denoted as $f_i$. During the standard training process, all layers learn useful features which contribute to the correct outputs of the model. We denote the hidden feature learned at the output of the $i$-th layer as $F_i(x) = f_i(f_{i-1}(\ldots f_1(x)))$.

Under adversarial perturbation, features from all layers will be altered, leading to incorrect outputs. Following previous work (Hein & Andriushchenko, 2017; Finlay et al., 2018), the robustness, or weakness, of the feature can be linked to the local Lipschitz constant of function $F_i(\cdot)$. For a easier computation, we consider the worst-case feature difference under a fixed input perturbation budget of $\epsilon$. This leads to our definition of the $\epsilon$-weakness of layer $i$'s feature as:

$$\mathcal{W}_\epsilon(F_i) = \frac{1}{N_i} \mathbb{E}_x \left[ \sup_{||\delta||_p \leq \epsilon} ||F_i(x + \delta) - F_i(x)||_2 \right], \tag{2}$$

where $N_i$ denotes the dimensionality of the output features at layer $i$, therefore normalizing the weakness measurement of layers with different output sizes. The weakness measurement is proportional to the local Lipschitz constant. A higher weakness value indicates that the feature vector is more vulnerable to input perturbations. The functionality of cascading layers from 1 to $i$ affects the vulnerability of the hidden features, as described by this formulation.

Alternatively, Moosavi-Dezfooli et al. (2019) suggests the local curvature can be a more accurate estimation of robustness than the Lipschitz constant. However, the computation and optimization of curvature involves costly higher-order gradient computation. We provide additional derivations in Appendix E to show the feature weakness defined in Eq. (2) is also an effective approximation to the local curvature value.

For the purpose of mitigating overfitting, we want to identify the layers that are the most critical to the lack of robustness, characterized by their increased susceptibility to perturbations from adversarial inputs. Other already-robust layers shall then be fixed to avoid further overfitting to the adversarial training objective. We therefore provide the following definition:

**Definition 3.1. Critical layer:** A layer is considered critical if it exhibits a greater propensity to learn non-robust features or demonstrates diminished robustness to adversarial input perturbations relative to other layers in the model.

To this end, we single out the contribution of each layer's functionality to the weakness of the features after it with a *Layer Criticality Index* $\mathcal{C}_{f_i}$, which is formulated as

$$\mathcal{C}_{f_i} = \frac{\mathcal{W}_\epsilon(F_i)}{\mathcal{W}_\epsilon(F_{i-1})}. \tag{3}$$

For the first layer, we define $\mathcal{C}_{f_1} = \mathcal{W}_\epsilon(F_1)$ as only the first layer contributes to the weakness.

As a sanity check, the feature weakness at the output of layer $i$ can be attributed to the criticality of all previous layers as $\mathcal{W}_\epsilon(F_i) = \prod_{k=1}^{i} \mathcal{C}_{f_k}$. Conversely, a layer with a larger criticality index will increase the weakness of the features after it, indicating the layer is critical according to Definition 3.1, as it weakens the feature robustness to the adversarial input.

One drawback of the formulation in Eq. (3) is that computing the feature weakness involves finding the worst-case perturbation against the hidden features at each layer, which is a costly process to conduct sequentially for all layers. In practice, we approximate the worst-case perturbation against features with an untargeted PGD attack against the model output, so that we can use the same PGD perturbation $\delta$ to estimate the feature weakness of all layers following Eq. (2). In this way, with a reasonably sufficient batch size, we can compute the critical indices for all layers in a model with two forward passes: one with the clean input $x$ and one with the PGD attack input $x + \delta$. We make the following proposition:

**Proposition 3.2.** *Critical layers defined as in Definition 3.1 can be identified as the layers with the largest criticality indices* $\arg\max_i \mathcal{C}_{f_i}$.

To verify Proposition 3.2, we conduct an ablation study in Table 7, where we show that model robustness is improved more by CLAT fine-tuning of critical layers compared to equivalent fine-tuning of randomly selected layers. We will discuss how to reduce the criticality of the critical layers and make them more robust in the next subsection.

## 3.2 CRITICALITY-TARGETED FINE-TUNING

Once the critical layers are identified, we fine-tune them to reduce their criticality, thereby decreasing the weakness of subsequent hidden features and enhancing model robustness. For a critical layer $i$, we optimize the trainable parameters to minimize $\mathcal{C}_{f_i}$. Note that in the criticality formulation in Eq. (3), the weakness of the previous layer's output, $\mathcal{W}_\epsilon(F_{i-1})$, is constant with respect to $f_i$. Thus, the optimization objective for $f_i$ can be simplified as

$$\mathcal{L}_C(f_i) = \mathbb{E}_x \left[ \sup_{||\delta||_p \leq \epsilon} ||F_i(x + \delta) - F_i(x)||_2 \right]. \tag{4}$$

In the case where multiple critical layers are considered in the fine-tuning process, the fine-tuning objective can be expanded to accommodate all critical layers simultaneously. Formally, suppose we have a set $S$ where layers $i \in S$ are all selected for fine-tuning, the fine-tuning objective for these critical layers can be formulated as

$$\mathcal{L}_C(f_S) = \mathbb{E}_x \left[ \sup_{||\delta||_p \leq \epsilon} \sum_{i \in S} ||F_i(x + \delta) - F_i(x)||_2 \right], \tag{5}$$

where a single perturbation is utilized to capture the weakness across all critical layers. In the common setting, a projected gradient ascent optimization with random start is used for the inner maximization.

Minimizing the objective in Eq. (5) by adjusting the trainable variables of the critical layers will reduce their feature weaknesses. However, the removal of non-robust features in these layers may affect the functionality of the model on clean inputs. As a tradeoff, we also include the cross entropy loss $\mathcal{L}(\cdot)$ in the final optimization objective, which derives the optimization objective on the critical layers during the fine-tuning process

$$\min_{f_S} \mathbb{E}_{x,y} \mathcal{L}(F(x), y) + \lambda \mathcal{L}_C(f_S), \tag{6}$$

where the hyperparameter $\lambda$ serves as a balancing factor between the two loss terms. Note that only the selected critical layers $f_S$ are optimized in Eq. (6) while the other non-critical layers are frozen, preventing them from further overfitting.

## 3.3 CLAT ADVERSARIAL TRAINING

We design CLAT as a fine-tuning approach, which is applied to neural networks that have undergone preliminary training. The pretraining phase allows all layers in the model to capture useful features, which will facilitate the identification of critical layers in the model. Notably, CLAT's versatility allows it to adapt to various types of pretrained models, either adversarially trained or trained on a clean dataset only. In practice, we find that models do not need to fully converge during the pretraining phase to benefit from CLAT fine-tuning. For example, in case of the CIFAR-10 dataset, 50 epochs of PGD-AT training would be adequate. We consider the number of adversarial pretraining epochs as a hyperparameter and provide further analysis on the impact of pretraining epochs in Section 4.3.1.

After the pretraining, CLAT begins by identifying and selecting critical layers in the pretrained model. We then fine-tune critical layers only while freezing the rest of the layers. This process is illustrated in Fig. 1, and the pseudocode is provided in Algorithm 1 in **Appendix A** for greater clarity. As fine-tuning progresses, the critical layers will be updated to reduce their criticality, making them less critical than some of the previously frozen layers. Subsequently, we perform periodic reevaluation of the top $k$ critical indices, ensuring continuous adaptation and optimization of the layers that are the most in need in the training process. Through hyperparameter optimization, we find 10 epochs to be adequate to optimize the selected critical layers for all models that we tested.

## 4 EXPERIMENTS

### 4.1 EXPERIMENTAL SETTINGS

**Datasets and models**  We conducted experiments using two widely recognized image classification datasets, CIFAR10 and CIFAR100. Each dataset includes 60,000 color images, each 32×32 pixels, divided into 10 and 100 classes respectively (Krizhevsky & Hinton, 2009). For our experiments, we deployed a suite of network architectures: WideResnets (34-10, 70-16) (Zagoruyko & Komodakis, 2017), ResNets (50, 18) (He et al., 2016b), DenseNet-121 (Huang et al., 2017), PreAct ResNet-18(He et al., 2016a), and VGG-19 (Simonyan & Zisserman, 2015). In this paper, these architectures are referred to as WRN34-10, WRN70-16, RN50, RN18, DN121, Preact RN18 and VGG19 respectively.

**Training and evaluation**  Since CLAT can be layered over clean pretraining, partial training, or other adversarial methods, results incorporating CLAT are denoted in our tables as "X + CLAT," where "X" refers to the baseline method applied prior to CLAT. Typically, this baseline method is run for the first 50 epochs, followed by fine-tuning during which CLAT is applied for an additional 50 epochs. The total number of epochs is in line with the 100 epochs used in previous PGD-based adversarial training work (Zhang et al., 2019; Zhu et al., 2023).

For our baseline, we use PGD for attack generation during training, following a random start (Madry et al., 2019), with an attack budget of $\epsilon = 0.03$ under the $\ell_\infty$ norm, a step size of $\alpha = 0.007$, and 10 attack steps. The same settings apply to PGD attack evaluations. AutoAttack evaluations (Croce & Hein, 2020) also use a budget of $\epsilon = 0.03$ under the $\ell_\infty$ norm, with no restarts for untargeted APGD-CE, 9 target classes for APGD-DLR, 3 target classes for Targeted FAB, and 5000 queries for Square Attack. These settings remain consistent unless explicitly noted otherwise.

Experiments were conducted on a Titan XP GPU, starting with an initial learning rate of 0.1, which was adjusted according to a cosine decay schedule. To ensure the reliability of robustness measurements, we conducted each experiment a minimum of 10 times, reporting the lowest adversarial accuracies we observed.

**CLAT settings**  We select critical layers as described in Section 3.1. Table 8 outlines the Top-5 most critical layers for some of the models and corresponding datasets at the start of the CLAT fine-tuning, after adversarially training with pgd-at for 50 epochs. In customizing the CLAT methodology to various network sizes, we select approximately 5% of layers as critical through hyperparameter optimization. For instance, DN121 uses 5 critical layers, while WRN70-16, RN50, WRN34-10, VGG19, and RN18 use 4, 3, 2, 1, and 1 critical layers, respectively.

## 4.2 CLAT PERFORMANCE

**White-box robustness**    Table 1 and Table 2 present white-box evaluation results using the PGD and Auto attack frameworks, respectively. These tables illustrate CLAT's versatility and effectiveness when combined with various standard adversarial training methods, including state-of-the-art benchmarks from RobustBench (Croce et al., 2020). CLAT effectively mitigates the overfitting typically observed in traditional adversarial training, thereby enhancing both clean and adversarial accuracy compared to baseline methods.

Table 1 further highlights that reducing trainable parameters alone does not necessarily lead to improved performance. CLAT surpasses other parameter-efficient fine-tuning methods, such as LoRA (Aleem et al., 2024) and RiFT (Zhu et al., 2023), due to its ability to precisely identify critical layers and eliminate non-robust features from these layers. Additionally, we show that fast adversarial training techniques, as discussed by Wong et al. (2020), can be applied to address the inner maximization problem in the CLAT training objective described in Eq. (5). The "CLAT (Fast)" method not only enhances performance but also improves robustness compared to Fast-AT baselines.

Notably, CLAT models are trained with PGD-like attacks on hidden features without seeing Auto Attacks directly, but their robustness persists under these attacks (see Table 2). This suggests that different attacks across various networks share similarities in exploiting non-robust features. By addressing these non-robust features through critical layer fine-tuning, CLAT ensures that the robustness is adaptable across different attack settings and models. Lastly, we assess the robustness across various attack strengths in **Appendix B**.

**Black-box robustness**    Table 3 and Table 4 evaluate the robustness against black-box attacks (Auto Attack and PGD-AT respectively) between models trained solely using PGD-AT and those augmented with CLAT. Attack settings are the same as those of the white-box attacks. As a sanity check, the accuracies under black-box attack surpass those observed under white-box scenarios, indicating that gradient masking does not appear in the CLAT model, and that the white-box robustness evaluation is valid. More significantly, models trained with CLAT consistently outperform those trained with PGD-AT, maintaining superior resilience in both black-box and white-box settings, regardless of the attack method or models employed.

## 4.3 ABLATION STUDIES

### 4.3.1 ABLATING ON PRETRAINING EPOCHS BEFORE CLAT

As discussed in Section 3.3, we apply CLAT after the model has been adversarially trained for some epochs. Here, we analyze how the number of pretraining epochs affects CLAT performance. Fig. 2 shows the training curves for different allocations of PGD pretraining epochs and CLAT fine-tuning epochs within a 100-epoch training budget. The overfitting of PGD-AT is evident as adversarial accuracy plateaus and declines towards the end, as documented in previous research (Rice et al., 2020). In contrast, CLAT continues to improve adversarial accuracy, effectively addressing this issue. Including CLAT at any stage of training results in higher clean accuracy and robustness at convergence. Additional results on pretrained clean models are provided in **Appendix C**.

Furthermore, an intriguing aspect of our experiments involves running CLAT from scratch (0 PGD-AT epochs). Although CLAT ultimately surpasses PGD-AT with sufficient epochs, using CLAT without any prior adversarial training results in significantly slower model convergence. We believe this suggests that "layer criticality" emerges during the adversarial training process, allowing critical layers to be identified as the model undergoes adversarial training. This phenomenon supports our theoretical insight that criticality can be linked to the curvature of the local minima to which each layer converges during adversarial training.

### 4.3.2 ABLATING ON CRITICAL LAYER SELECTION

The choice of critical layer selection is another important feature impacting the performance of CLAT. We begin by examining the effect of dynamic layer selection. Table 5 and Table 6 highlight that dynamic selection is crucial to CLAT's performance. Using the same layers throughout the process tends to cause overfitting and results in lower accuracies compared to the PGD-AT baseline.

Table 1: Comparative performance of CLAT across various networks and adversarial training/fine-tuning techniques. Robustness is evaluated with white-box PGD attacks.

| MODEL | METHOD | CIFAR-10 ACC. (%) | | CIFAR-100 ACC. (%) | |
|---|---|---|---|---|---|
| | | CLEAN | ADVERSARIAL | CLEAN | ADVERSARIAL |
| DN121 | PGD-AT (MADRY ET AL., 2019) | 80.05 | 58.15 | 57.18 | 31.76 |
| | **PGD-AT + CLAT** | **81.03** | **60.60** | **58.79** | **33.23** |
| WRN70-16 | (PENG ET AL., 2023) | 93.27 | 71.07 | 70.20 | 42.61 |
| | **PENG ET AL. + CLAT** | **93.56** | **72.25** | **71.94** | **44.12** |
| | (BAI ET AL., 2024) | 92.23 | 64.55 | 69.17 | 40.86 |
| | **BAI ET AL. + CLAT** | **92.77** | **64.92** | **70.17** | **41.64** |
| RN50 | PGD-AT | 81.38 | 56.35 | 58.16 | 33.01 |
| | **PGD-AT + CLAT** | **83.78** | **59.54** | **61.88** | **36.23** |
| WRN34-10 | PGD-AT | 87.41 | 55.40 | 59.19 | 31.66 |
| | PGD-AT + LORA (ALEEM ET AL., 2024) | 73.36 | 56.17 | 55.56 | 31.43 |
| | PGD-AT + RIFT (ZHU ET AL., 2023) | 87.89 | 55.41 | 62.35 | 31.64 |
| | **PGD-AT + CLAT** | **88.97** | **57.11** | **62.38** | **32.05** |
| | TRADES (ZHANG ET AL., 2019) | 87.60 | 56.61 | 60.56 | 31.85 |
| | TRADES + RIFT | 87.55 | 56.72 | 61.01 | 32.03 |
| | **TRADES + CLAT** | **88.23** | **57.89** | **61.45** | **33.56** |
| VGG19 | PGD-AT | 78.38 | 50.35 | 50.16 | 26.54 |
| | **PGD-AT + CLAT** | **79.88** | **52.54** | **50.98** | **28.41** |
| RN18 | PGD-AT | 81.46 | 53.63 | 57.10 | 30.15 |
| | PGD-AT + LORA | 76.57 | 55.38 | 48.49 | 32.36 |
| | PGD-AT + RIFT | 83.44 | 53.65 | 58.74 | 30.17 |
| | **PGD-AT + CLAT** | **83.89** | **55.37** | **59.22** | **32.04** |
| | TRADES | 81.54 | 53.31 | 57.44 | 30.20 |
| | TRADES + RIFT | 81.87 | 53.30 | 57.78 | 30.22 |
| | **TRADES + CLAT** | **81.89** | **54.57** | **58.82** | **31.06** |
| | MART (WANG ET AL., 2019) | 76.77 | 56.90 | 51.46 | 31.47 |
| | MART + RIFT | 77.14 | 56.92 | 52.42 | 31.48 |
| | **MART + CLAT** | **76.82** | **57.55** | **53.01** | **33.23** |
| | AWP (WU ET AL., 2020B) | 78.40 | 53.83 | 52.85 | 31.00 |
| | AWP + RIFT | 78.79 | 53.84 | 54.89 | 31.05 |
| | **AWP + CLAT** | **79.01** | **55.27** | **55.39** | **32.08** |
| | SCORE (PANG ET AL., 2022) | 84.20 | 54.59 | 54.83 | 29.49 |
| | SCORE + RIFT | 85.65 | 54.62 | 57.63 | 29.50 |
| | **SCORE + CLAT** | **86.11** | **55.78** | **57.66** | **30.23** |
| | AUTOLORA (ZHANG ET AL., 2024) | 84.2 | 54.27 | 62.1 | 32.71 |
| | **AUTOLORA + CLAT** | **86.45** | **58.53** | **64.91** | **35.82** |
| PREACT RN18 | FAST-AT (WONG ET AL., 2020) | 81.46 | 45.55 | 50.10 | 27.72 |
| | **FAST-AT + CLAT** | **84.46** | **52.13** | **54.33** | **29.22** |
| | **FAST-AT + CLAT (FAST)** | **82.72** | **49.62** | **52.10** | **27.99** |

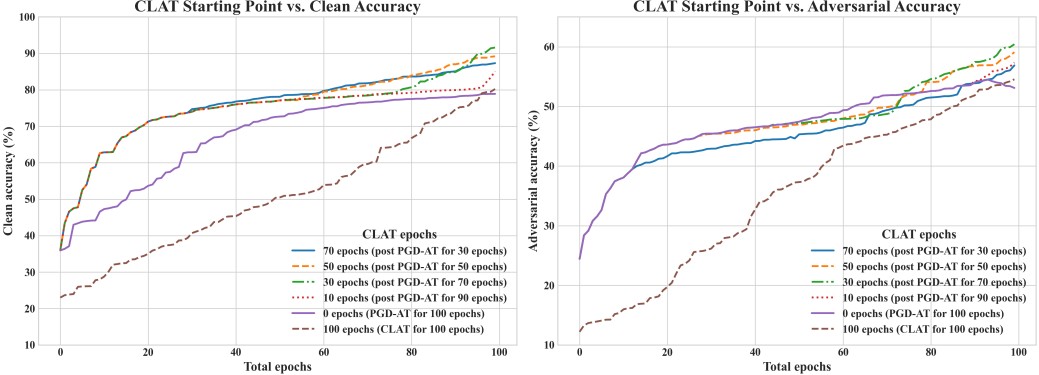

Figure 2: Comparative analysis of CLAT performance on WRN34-10: Clean and adversarial accuracy on CIFAR-10 across partially trained models.

To verify the significance of the selected critical layers, we compare CLAT with an alternative approach in which random layers are dynamically selected for fine-tuning instead of the critical layers. The results of this comparison are detailed in Table 7 and Table 12 (in **Appendix C**).

Table 2: Adversarial Accuracy on CIFAR-10 and CIFAR-100 when subjected to AutoAttack (AA).

| Network | Method | Adversarial Accuracy (%) | |
|---|---|---|---|
| | | CIFAR-10 | CIFAR-100 |
| DN121 | PGD-AT | 47.56 | 23.13 |
| | **PGD-AT + CLAT** | **49.91** | **25.74** |
| WRN70-16 | PGD-AT | 54.32 | 28.25 |
| | **PGD-AT + CLAT** | **57.64** | **30.98** |
| | (Carlini & Wagner, 2017) | 66.1 | - |
| | **(Carlini & Wagner, 2017) + CLAT** | **68.43** | - |
| RN50 | PGD-AT | 46.22 | 23.48 |
| | **PGD-AT + CLAT** | **49.45** | **25.81** |
| | (Madry et al., 2019) | 49.25 | - |
| | **(Madry et al., 2019) + CLAT** | **51.64** | - |
| WRN34-10 | PGD-AT | 51.50 | 25.56 |
| | **PGD-AT + CLAT** | **52.88** | **27.62** |
| | (Croce & Hein, 2020) | 57.3 | - |
| | **(Croce & Hein, 2020) + CLAT** | **59.98** | - |
| VGG19 | PGD-AT | 40.42 | 19.54 |
| | **PGD-AT + CLAT** | **41.72** | **20.45** |
| RN18 | PGD-AT | 40.48 | 20.21 |
| | **PGD-AT + CLAT** | **42.86** | **21.76** |
| | Twins (Liu et al., 2023) | 47.89 | 25.45 |
| | **Twins + CLAT** | **51.39** | **28.12** |
| | AutoLoRA (Zhang et al., 2024) | 48.95 | 27.48 |
| | **AutoLoRA + CLAT** | **53.21** | **30.49** |

Table 3: Comparative Analysis of Black-box Auto Attack (AA) Accuracy on CIFAR-10 and CIFAR-100. Model in each row is the attacker and each column the victim.

| Network | Method | CIFAR-10 Adv. Acc. (%) | | | | CIFAR-100 Adv. Acc. (%) | | | |
|---|---|---|---|---|---|---|---|---|---|
| | | DN121 | RN50 | VGG19 | RN18 | DN121 | RN50 | VGG19 | RN18 |
| DN121 | PGD-AT | - | 52.50 | 44.21 | 45.45 | - | 27.64 | 23.21 | 24.26 |
| | PGD-AT + CLAT | - | 55.83 | 47.53 | 48.92 | - | 29.89 | 26.71 | 26.93 |
| RN50 | PGD-AT | 54.23 | - | 43.56 | 43.24 | 27.91 | - | 23.02 | 23.51 |
| | PGD-AT + CLAT | 56.72 | - | 46.21 | 47.01 | 30.11 | - | 26.34 | 25.86 |
| VGG19 | PGD-AT | 55.32 | 55.45 | - | 46.72 | 27.84 | 28.15 | - | 24.20 |
| | PGD-AT + CLAT | 59.83 | 59.72 | - | 49.31 | 30.20 | 29.79 | - | 26.55 |
| RN18 | PGD-AT | 53.21 | 51.73 | 43.21 | - | 29.31 | 26.75 | 22.91 | - |
| | PGD-AT + CLAT | 56.75 | 54.45 | 46.53 | - | 31.25 | 29.41 | 25.95 | - |

The data demonstrates that selecting critical layers significantly enhances the model's adversarial robustness and clean accuracy. This observation is bolstered by our ablation study in **Appendix Table 20** illustrating the performance effect of choosing the smallest versus largest critical indices for fine-tuning. Furthermore, Table 8 indicates near-identical critical layer selections within the same model, even across diverse datasets. This evidence supports our assertion that the variation in layer criticality arises from inherent properties within the model architecture, where certain layers are predisposed to learning non-robust features.

Lastly, we conduct an ablation study on the number of layers used in CLAT for fine-tuning. Fig. 3 and Fig. 7 illustrate the trade-offs between adversarial accuracy and the number of layers selected, as well as clean accuracy and the number of layers, respectively. Interestingly, both adversarial and clean accuracy are optimized with the same number of layers. Initially, fine-tuning more layers enhances model performance by increasing flexibility; however, this eventually diminishes CLAT's effectiveness, likely because attention is diverted to less crucial layers at the expense of more important ones. This pattern underscores the critical role of specific layers in network robustness and emphasizes the need for deeper research into the dynamics of individual layers. Furthermore,

Table 4: Comparative Analysis of Black-box PGD Accuracy on CIFAR-10 and CIFAR100. Model in each row is the attacker and each column the victim.

| Network | Method | Cifar-10 Adv. Acc. (%) | | | | Cifar-100 Adv. Acc. (%) | | | |
|---|---|---|---|---|---|---|---|---|---|
| | | RN50 | DN121 | VGG19 | RN18 | RN50 | DN121 | VGG19 | RN18 |
| RN50 | PGD-AT | - | 74.83 | 68.01 | 67.44 | - | 46.82 | 40.55 | 40.10 |
| | PGD-AT + CLAT | - | **76.45** | **71.25** | **70.12** | - | **48.49** | **44.34** | **43.91** |
| DN121 | PGD-AT | 72.24 | - | 69.53 | 68.38 | 44.45 | - | 40.63 | 41.22 |
| | PGD-AT + CLAT | **74.55** | - | **71.78** | **70.56** | **46.78** | - | **43.62** | **42.88** |
| VGG19 | PGD-AT | 65.72 | 67.56 | - | 62.26 | 47.86 | 46.56 | - | 40.55 |
| | PGD-AT + CLAT | **66.46** | **70.72** | - | **65.78** | **49.25** | **48.72** | - | **42.72** |
| RN18 | PGD-AT | 74.82 | 70.21 | 61.83 | - | 46.28 | 45.59 | 39.21 | - |
| | PGD-AT + CLAT | **76.23** | **73.19** | **63.96** | - | **48.89** | **47.72** | **41.78** | - |

Table 5: Adversarial accuracy under PGD attack: Comparison of PGD-AT, PGD-AT + CLAT, and PGD-AT + CLAT (Non-dynamic) on CIFAR-10 and CIFAR-100.

Table 6: Adversarial accuracy under Auto attack: Comparison of PGD-AT, PGD-AT + CLAT, and PGD-AT + CLAT (Non-dynamic) on CIFAR-10 and CIFAR-100.

| Method | CIFAR-10 | | CIFAR-100 | |
|---|---|---|---|---|
| | DN121 | RN50 | DN121 | RN50 |
| PGD-AT | 58.15 | 56.35 | 31.76 | 33.01 |
| CLAT | **60.60** | **59.54** | **33.23** | **36.23** |
| CLAT (ND) | 57.01 | 54.22 | 30.34 | 32.98 |

| Method | CIFAR-10 | | CIFAR-100 | |
|---|---|---|---|---|
| | DN121 | RN50 | DN121 | RN50 |
| PGD-AT | 47.56 | 46.22 | 23.13 | 23.48 |
| CLAT | **49.91** | **49.45** | **25.74** | **25.81** |
| CLAT (ND) | 47.12 | 46.08 | 22.26 | 22.91 |

we highlight that although the number of layers chosen impacts the learned robustness, CLAT still achieves robustness gains over the early-stopping baseline (no fine-tuning) with up to 10% of the layers selected, demonstrating its stability under small variations in the number of selected layers.

## 4.4 Overhead and stability analysis

The cost of optimizing the CLAT training objective in Eq. (6) is similar to that of the standard adversarial training given its min-max formulation. Here, we show that the computational overhead for determining critical indices is negligible. We verify in Table 10 that criticality indices can be stably computed with a single training batch as small as 10, with top-ranking layers consistent with those achieved with a larger batch. We further conducted over 1000 runs for each network to randomly select the data used for criticality estimation, where we find remarkable consistency in computed critical layers, differing in less than 5% of cases, typically involving only one layer change among the top five critical layers. The stability means we can use a mere **0.0002%** of the training data for criticality estimation every 10 epochs, which introduces a neglectable **0.4%** additional time to the standard adversarial training process as measured in Table 10.

Table 7: Ablating CLAT Layer choices on CIFAR-10: The columns present PGD-10 and Auto Attack (AA) evaluation adversarial accuracies for models trained with CLAT, where layers are selected based on criticality or randomly.

| Network | Critical Layers | | | Random Layers | | |
|---|---|---|---|---|---|---|
| | Clean Acc. | PGD-10 | AA | Clean Acc. | PGD-10 | AA |
| DN121 | 81.03 | 60.60 | 49.91 | 78.85 | 51.35 | 39.81 |
| RN50 | 83.78 | 59.54 | 49.45 | 79.01 | 51.44 | 40.29 |
| RN18 | 83.89 | 55.37 | 42.86 | 78.02 | 51.03 | 33.50 |

Table 8: Top-5 criticality indices by model and dataset. Layers used in CLAT are bolded.

| MODEL | CIFAR10 | CIFAR100 |
|---|---|---|
| DN121 | **39, 14, 1, 3, 88** | **39, 15, 1, 2, 91** |
| WRN70-16 | **4, 17, 1, 59**, 62 | **3, 17, 2, 59**, 61 |
| RN50 | **34, 41, 48**, 3, 36 | **34, 43, 45**, 6, 32 |
| WRN34-10 | **26, 1** 30, 3, 28 | **26, 2**, 30, 3, 27 |
| VGG19 | **9**, 11, 5, 3, 1 | **8**, 13, 5, 3, 1 |
| RN18 | **11**, 10, 4, 2, 12 | **12**, 9, 5, 2, 13 |

Table 9: Trainable Parameters during CLAT in Various Networks

| NETWORK | TRAINABLE PARAMS | | % USED |
|---|---|---|---|
| | TOTAL | CLAT | |
| DN121 | 6.96M | 217K | 3.1% |
| WRN70-16 | 267M | 8.29M | 3.0% |
| RN50 | 23.7M | 823K | 3.4% |
| WRN34-10 | 46.16M | 1.24M | 2.7% |
| RN18 | 11.2M | 590K | 5.2% |
| VGG19 | 39.3M | 236K | 0.6% |

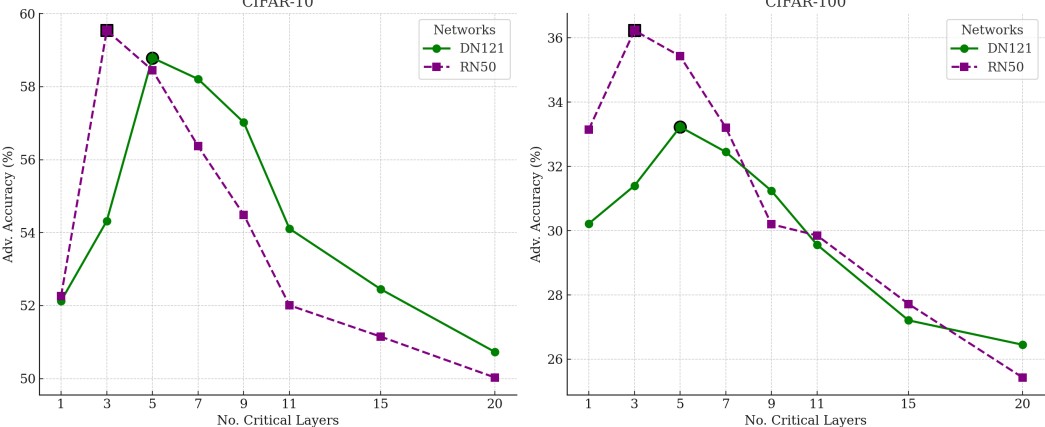

Figure 3: Comparative analysis on CLAT performance/PGD-10 adversarial accuracy with respect to number of critical layers used during CLAT

## 5 CONCLUSIONS

In this work, we introduce CLAT, an innovative adversarial training approach that addresses robust overfitting issues by fine-tuning only the critical layers vulnerable to adversarial perturbations. This method not only emphasizes layer-specific interventions for enhanced network robustness but also sheds light on the commonality in non-robust features captured by these layers, offering a targeted and effective defense strategy. Our results reveal that CLAT effectively selects less than 4% critical trainable variables to achieve significant improvements in clean accuracy and adversarial robustness across diverse network architectures and baseline adversarial training methods. We limit the scope of this work to improving the empirical robustness of the model by utilizing the critical layers. In this sense, open questions remain on why these specific layers become critical, whether they can be identified more effectively, and whether the issues can be resolved with architectural or training scheme changes. We leave a more theoretical understanding of these questions as future work.

Table 10: DenseNet-121 critical layers identified with different amount of data. Time taken to compute critical layers evaluated on TITAN RTX GPU. As a reference, 1 PGD-AT epoch takes 67s.

| BATCH SIZE | CIFAR10 | | CIFAR100 | |
|---|---|---|---|---|
| | CRITICAL LAYERS | TIME (S) | CRITICAL LAYERS | TIME(S) |
| 10 | 39, 14, 1, 3, 90 | 2.64 | 39, 15, 1, 2, 91 | 2.82 |
| 30 | 39, 14, 1, 3, 88 | 2.72 | 39, 15, 1, 2, 88 | 2.91 |
| 50 | 39, 14, 1, 3, 89 | 2.83 | 39, 15, 1, 3, 91 | 3.14 |
| 100 | 39, 14, 1, 3, 88 | 3.15 | 39, 15, 1, 2, 91 | 3.54 |

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

## A  PSEUDOCODE OF CLAT

To better facilitate an understanding of the CLAT process, we illustrate the pseudocode of the dynamic critical layer identification process and the criticality-targeted fine-tuning process in **Algorithm 1**. Only the selected critical layers are being fine-tuned while all the other layers are frozen.

---

**Algorithm 1** CLAT Algorithm

---

1: **Input:** Dataset $\mathcal{D}$, pre-trained model $F$, batch size $bs$, total epochs $N$.

2: **for** $epoch = 1$ **to** $N$ **do**
3:     **if** $epoch\%10 == 1$ **then**
4:         # Find critical layers
5:         $x \leftarrow$ Batch of training data in $\mathcal{D}$
6:         $x + \delta \leftarrow$ PGD attack against $F$
7:         Compute $\mathcal{W}_\epsilon(F_i)$ for all layers with Eq. (2)
8:         Compute $\mathcal{C}_{f_i}$ for all layers with Eq. (3)
9:         Critical layers $\mathcal{S} \leftarrow TopK(\mathcal{C}_{f_i})$
10:     # fine-tune critical layers
11:     $minibatches \leftarrow CreateMinibatches(\mathcal{D}, bs)$
12:     **for** $x, y$ in $minibatches$ **do**
13:         Perturbation $\delta \leftarrow$ Eq. (5) inner maximization
14:         $\mathcal{L}_C(f_\mathcal{S}) \leftarrow$ Eq. (5)
15:         Weight update $w[\mathcal{S}]$ with Eq. (6)

---

## B  ADDITIONAL EXPERIMENT RESULTS

We further compare the CLAT model robustness with the robustness of the SAT model against white-box attacks of various strengths. As illustrated in Fig. 4, though both models are only trained against an attack of one strength ($\epsilon = 0.03$), the improved robustness of CLAT is consistent across the full spectrum of attack strengths. This shows that CLAT is not overfitting to the specific attack strength used in training.

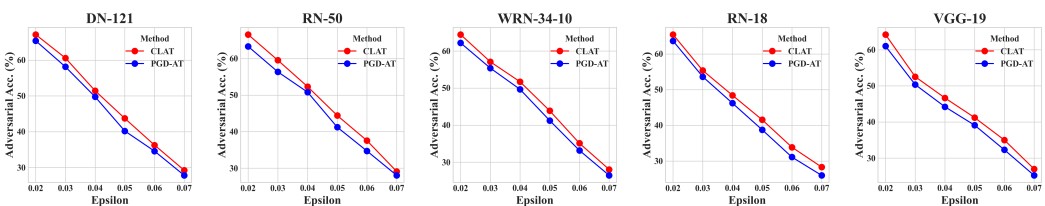

Figure 4: White-box adversarial accuracy (y-axis) on CIFAR-10 for models trained with CLAT (red) and pgd-at (blue), against PGD attacks of varying strengths (x-axis)

## C  ADDITIONAL ABLATION STUDY

### C.1  CLAT ON PRETRAINED CLEAN MODEL

Besides the discussion on performing CLAT after adversarial pretraining in Section 4.3.1, Table 11 details the performance of CLAT on clean pretrained models. Although the adversarial accuracies of clean pretrained models are relatively low compared to those of adversarially trained models, CLAT demonstrates its capability to facilitate adversarial fine-tuning on clean models effectively to some extent. This is a novel achievement, showcasing the algorithm's versatility.

Table 11: Adversarial and clean accuracies for performing CLAT on various PyTorch pretrained models on the CIFAR-10 dataset.

| MODEL | ADV. ACC. | CLEAN ACC. |
|---|---|---|
| DN-121 | 39.21% | 80.89% |
| WRN70-16 | 42.1% | 83.35% |
| RN-50 | 35.67% | 78.23% |
| WRN34-10 | 40.1% | 81.78% |
| VGG-19 | 32.67% | 75.05% |
| RN-18 | 34.45% | 76.51% |

## C.2 ADDITIONAL RESULTS ON LAYER SELECTION

Here in Table 12, we provide additional results contrasting CLAT with an alternative approach where random layers are dynamically selected for fine-tuning instead of the critical layers on the CIFAR-100 dataset.

Table 12: Ablating CLAT layer choices on CIFAR-100: The columns present PGD-10 and AutoAttack (AA) evaluation adversarial accuracies for models trained with CLAT, where layers are selected based on criticality or randomly.

| NETWORK | CRITICAL LAYERS | | | RANDOM LAYERS | | |
|---|---|---|---|---|---|---|
| | CLEAN ACC. | PGD-10 | AA | CLEAN ACC. | PGD-10 | AA |
| DN121 | 58.79 | 33.23 | 25.74 | 50.45 | 25.48 | 20.21 |
| RN50 | 61.88 | 36.23 | 25.81 | 52.20 | 25.51 | 20.45 |
| RN18 | 50.98 | 28.41 | 20.45 | 43.15 | 20.24 | 15.89 |

## D REBUTTAL EXPERIMENTS

We thank the constructive feedback from all reviewers. We consolidate all additional experimental results suggested by the reviewers in this appendix section. We will integrate these results and our discussions in the rebuttal text into the finalized version.

### D.1 FULL TRAINING CURVES

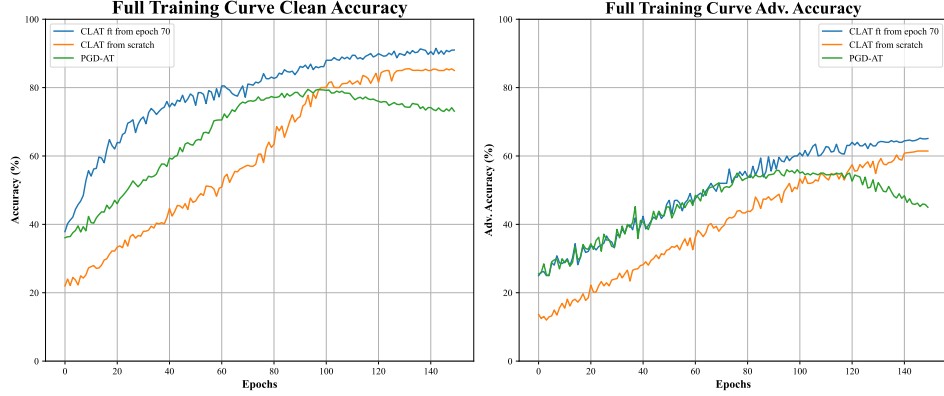

Figure 5: White-box PGD-10 adversarial accuracy (y-axis) on CIFAR-10 for WRN34-10 models trained with CLAT fine-tuning starting at Epoch 70 (red), CLAT from scratch (orange), and PGD-AT (blue). The learning rate decays to 0 by Epoch 150.

## D.2   REDUCED LEARNING RATE PERFORMANCE

Please see Fig. 6.

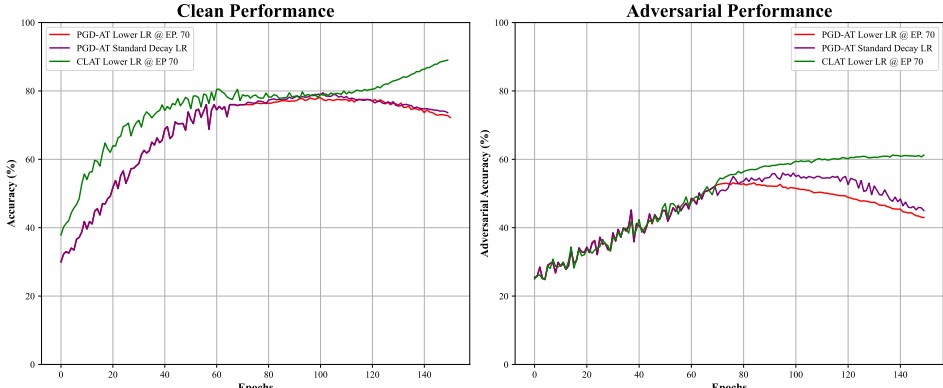

Figure 6: White-box accuracies (y-axis) for WRN34-10 on CIFAR-10 for models trained with the original learning rate multiplied by 0.1 at Epoch 70, using CLAT (green) and PGD-AT (red), compared to normal PGD-AT learning rate (purple).

## D.3   CRITICAL INDEX VARIATION OVER TIME

Please see Table 13.

Table 13: Critical layers identified at different epochs for various networks.

| NETWORK | EP. 70 | EP. 80 | EP. 90 |
|---|---|---|---|
| DN121 | [39, 14, 1, 3, 88] | [38, 1, 5, 88, 15] | [1, 5, 88, 2, 15] |
| RN50 | [34, 41, 48] | [48, 3, 36] | [36, 2, 40] |
| RN18 | [11] | [11] | [4] |

## D.4   CLEAN ACCURACY AND CRITICAL LAYER SELECTION

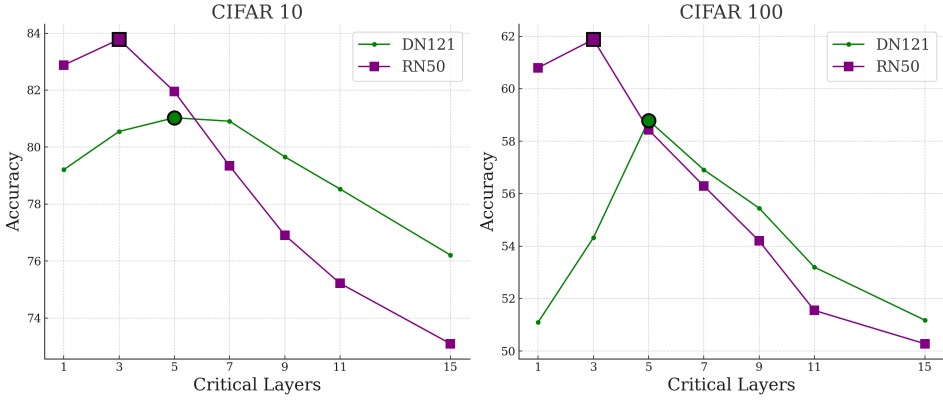

Figure 7: Comparative analysis on CLAT performance clean accuracies with respect to the number of critical layers used during CLAT.

## D.5   ABLATION STUDY: DYNAMIC AND FIXED CRITICAL INDICES

Please see Table 14.

Table 14: Comparison of CIFAR-100 Clean and Adversarial Accuracies (PGD-10) on Different Networks with Fixed and Dynamic Layers for CLAT.

| Model | | Fixed Layers | Dynamic Layers |
|-------|-------|-------|-------|
| DN121 | Clean | 53.21 | **58.79** |
|       | Adv   | 39.45 | **44.12** |
| RN50  | Clean | 55.72 | **61.88** |
|       | Adv   | 31.89 | **36.23** |
| WRN34-10 | Clean | 56.45 | **62.38** |
|          | Adv   | 28.56 | **32.05** |

## D.6 PERFORMANCE ON OTHER DATASETS

### D.6.1 IMAGENETTE

Results in Table 15.

Table 15: Comparative performance of CLAT across various networks on Imagenette. Robustness is evaluated with white-box PGD-10 and Auto Attack.

| Model | Method | Clean (%) | PGD-10 (%) | AA (%) |
|-------|--------|-----------|------------|--------|
| DN121 | PGD-AT | 83.40 | 61.78 | 51.23 |
|       | **PGD-AT + CLAT** | **86.91** | **65.45** | **54.82** |
| WRN70-16 | PGD-AT | 90.20 | 67.96 | 58.91 |
|          | **PGD-AT + CLAT** | **93.52** | **72.39** | **61.43** |
| RN50 | PGD-AT | 84.02 | 62.10 | 50.45 |
|      | **PGD-AT + CLAT** | **87.11** | **64.89** | **54.31** |
| WRN34-10 | PGD-AT | 90.45 | 65.45 | 56.31 |
|          | **PGD-AT + CLAT** | **93.21** | **69.82** | **60.04** |
| VGG19 | PGD-AT | 85.45 | 56.71 | 46.53 |
|       | **PGD-AT + CLAT** | **89.72** | **59.45** | **51.22** |
| RN18 | PGD-AT | 83.01 | 60.04 | 49.01 |
|      | **PGD-AT + CLAT** | **86.42** | **62.91** | **51.23** |

### D.6.2 IMAGENET

Results in Table 16.

Table 16: Clean and Adversarial Accuracies (PGD-10) performance comparison on ImageNet.

| Model | Method | Clean | Adv |
|-------|--------|-------|-----|
| DN121 | PGD-AT | 63.25 | 32.56 |
|       | **PGD-AT + CLAT** | **66.10** | **35.48** |
| RN50 | PGD-AT | 65.88 | 33.18 |
|      | **PGD-AT + CLAT** | **67.12** | **36.91** |
| WRN34-10 | PGD-AT | 64.31 | 31.08 |
|          | **PGD-AT + CLAT** | **66.12** | **33.59** |

## D.7 ROBUSTNESS AGAINST OTHER ATTACKS

We report that performance on varying strengths of PGD-10 in Appendix B. Table 17 highlights the robustness against stronger white-box attacks, including those not limited to $\ell_\infty$-bounded constraints.

Table 17: Adversarial accuracies across various attacks on CIFAR-10, highlighting the difference between models trained with the baseline and those utilizing CLAT. Positive blue values indicate the improvement in performance achieved with CLAT.

| Attack | Method | DN121 (%) | WRN34-10 (%) |
|---|---|---|---|
| FAB | PGD-AT | 44.80 | 40.12 |
| | PGD-AT + CLAT | +3.70 | +5.03 |
| StAdv | PGD-AT | 48.50 | 45.15 |
| | PGD-AT + CLAT | +0.91 | +1.89 |
| Pixle | PGD-AT | 10.40 | 9.50 |
| | PGD-AT + CLAT | +2.21 | +1.90 |
| PGD-$\ell_2$ ($\epsilon = 0.03$) | PGD-AT | 61.79 | 60.25 |
| | PGD-AT + CLAT | +1.92 | +1.45 |
| PGD-$\ell_\infty$ ($\epsilon = 0.03$, 50 steps) | PGD-AT | 57.01 | 54.01 |
| | PGD-AT + CLAT | +1.92 | +2.03 |
| PGD-$\ell_\infty$ ($\epsilon = 0.03$, 100 steps) | PGD-AT | 56.89 | 53.12 |
| | PGD-AT + CLAT | +1.63 | +1.89 |

## D.8 ADDITIONAL PERFORMANCE COMPARISONS TO BASELINES

### D.8.1 STOCHASTIC WEIGHT AVERAGING AND ADVERSARIAL WEIGHT PERTURBATION

Table 18 shows results when CLAT is augmented with SWA (Hwang et al., 2020) and AWP (Wu et al., 2020a) techniques. Omitted values are not reported in the original work.

Table 18: PGD-10 Adversarial Accuracies on CIFAR-10 and CIFAR-100 for PreAct RN-18 and WRN34-10 compared to baselines.

| Network | Method | CIFAR-10 | CIFAR-100 |
|---|---|---|---|
| PreAct RN-18 | AWP (Wu et al., 2020a) | 55.39 | 30.71 |
| | **AWP + CLAT** | **58.41** | **33.97** |
| | SWAAT (Hwang et al., 2020) | 58.32 | 28.43 |
| | **SWAAT + CLAT** | **60.76** | **30.74** |
| WRN34-10 | AWP | 58.10 | - |
| | **AWP + CLAT** | **60.89** | - |
| | SWAAT | 61.45 | 31.97 |
| | **SWAAT + CLAT** | **63.82** | **34.55** |

### D.8.2 DATA AUGMENTATION TECHNIQUES

Please see Table 19.

Table 19: Adversarial accuracies on CIFAR-10 for PreAct RN-18 (using PGD-10) and WRN70-16 (using Auto Attack).

| Network | Method | Adv. Accuracy (%) |
|---|---|---|
| PreAct RN-18 | Data Aug (weak) (Li & Spratling, 2023) | 50.34 |
| | **Data Aug (weak) + CLAT** | **54.01** |
| | Data Aug (strong) | 49.99 |
| | **Data Aug (strong) + CLAT** | **52.37** |
| WRN70-16 | Wang et al. (2023) | 70.69 |
| | **Wang et al. (2023) + CLAT** | **72.34** |

### D.9 Timing Comparisons

For a fair comparison, we use the same GPU configuration and number of GPUs across all methods, as described in the methods section. For DN121, one epoch of PGD-AT takes 67 seconds, RiFT takes 56 seconds per epoch, CLAT takes 69 seconds per epoch, and AutoLoRA also takes 69 seconds per epoch. As performed in the original paper for optimal performance, RiFT models were adversarially trained for 110 epochs, each taking 67 seconds, followed by fine-tuning for an additional 10 epochs at 56 seconds per epoch. Consequently, the total training time for RiFT is 132 minutes, compared to 112 minutes for CLAT (70 epochs of adversarial training and 30 epochs of fine-tuning).

### D.10 Ablation: Effect of choosing largest critical indices/ most critical layers

Please see Table 20.

Table 20: Ablation study of CLAT layer choices on CIFAR-10: The columns present Clean, PGD-10, and Auto Attack (AA) evaluation accuracies for models trained with CLAT, where layers with the largest and lowest values are selected. The optimal number of layers per network was chosen for both approaches. All settings are consistent with the results in Table 1

| Network | Largest cidx | | | Smallest cidx | | | PGD-AT | | |
|---|---|---|---|---|---|---|---|---|---|
| | Clean | PGD-10 | AA | clean | PGD-10 | AA | clean | PGD-10 | AA |
| DN121 | **81.03** | **60.60** | **49.91** | 80.50 | 59.25 | 48.81 | 80.05 | 58.15 | 47.56 |
| RN50 | **83.78** | **59.54** | **49.45** | 82.30 | 57.01 | 47.10 | 81.38 | 56.35 | 46.22 |
| RN18 | **83.89** | **55.37** | **42.86** | 82.56 | 54.01 | 40.91 | 81.46 | 53.63 | 40.48 |

## E Curvature-based weakness measurement

The main paper defines feature weakness based on the feature variation under worst-case perturbation. However, due to the non-linear and non-convex nature of the neural network model, the weakness measurement may not be precise in more complicated model architectures with a mixed layer type, such as the Vision Transformer model. To this end, this section provides a more accurate curvature-based formulation on feature weakness, and shows how the proposed weakness metric is an approximation. We leave the utilization of the curvature-based weakness measurement on more complicated models as future work.

Let's start by considering the feature perturbation function $G_i(\cdot)$, which is defined at the output of layer $i$ on inputs close to a clean data point $x$:

$$G_i(z) = ||F_i(z) - F_i(x)||_2^2. \tag{7}$$

The worst-case curvature of the function $G_i$ at the neighborhood of $z = x$ can be estimated following the formulation by Moosavi-Dezfooli et al. (2019) as

$$\nu_i(x) = \frac{\nabla G_i(x') - \nabla G_i(x)}{||x' - x||_2} = \frac{\nabla G_i(x')}{||x' - x||_2}, \tag{8}$$

where $x'$ is a worst-case perturbation (adversarial attack) maximizing $G_i(z)$ in the vicinity of $x$, and $\nabla G_i(x) = 0$ by definition given it is a minimum. Following the observation in Moosavi-Dezfooli et al. (2019), a higher curvature indicates the feature to be more non-robust to adversarial examples. We can therefore use the curvature formulation $\nu_i(x)$ under a fixed perturbation budget $||x' - x||_p \le \epsilon$ to estimate the layer non-robustness, or weakness.

As we use the feature weakness to derive both the layer criticality metric and the finetuning objective, having a gradient term in the layer weakness leads to the costly computation of higher-order gradients in the optimization. To avoid the high cost of computing higher-order gradients when optimizing with the curvature, the numerator in the curvature formulation can be further derived as

$$\nu_i(x) = \frac{\frac{\partial F_i(x')}{\partial x'}^T (F_i(x') - F_i(x))}{||x' - x||_2}. \tag{9}$$

In practice, it is also difficult to explicitly instantiate $\frac{\partial F_i(x')}{\partial x'}$ for a neural network. To this end, we simplify the formulation in Eq. (9) by assuming $\frac{\partial F_i(x')}{\partial x'}$ as a uniform vector. This leads to our definition of the $\epsilon$-weakness of layer $i$'s feature as:

$$\mathcal{W}_\epsilon(F_i) = \frac{1}{N_i}\mathbb{E}_x \left[ \sup_{||\delta||_p \leq \epsilon} ||F_i(x+\delta) - F_i(x)||_2 \right], \tag{10}$$

where $N_i$ denotes the dimensionality of the output features at layer $i$, therefore normalizing the weakness measurement of layers with different output sizes. The weakness measurement is proportional to the curvature estimation in Eq. (9). A higher weakness value indicates that the feature vector is more vulnerable to input perturbations. The functionality of cascading layers from 1 to $i$ affects the vulnerability of the hidden features, as described by this formulation.

