# OpenReview forum: "Boosting Adversarial Robustness with CLAT: Criticality Leveraged Adversarial Training"
_ICLR.cc/2025/Conference — Submitted to ICLR 2025_

### Official Review · Reviewer_oD5s · 2024-10-29

**Soundness:** 3
**Presentation:** 2
**Contribution:** 2
**Rating:** 6
**Confidence:** 4

**Summary:**

This work studies the adversarial robustness of image classification models. It focuses on improving adversarial robustness through fine-tuning the parameters of selected layers. Specifically, it first proposes to identify robustness-critical layers by the (approximate) curvature of loss landscape. During fine-tuning, it only updates the parameters in the robustness-critical layers, while keeping the rest fixed.

**Strengths:**

1. The idea of identifying and fine-tuning robustness-critical layers is interesting and less studied before. It has the potential of improving the trade-off between accuracy and robustness through reducing the over-regularization of adversarial training.

**Weaknesses:**

1. The paper uses the term "adversarial overfitting" without a clear definition. It is confusing because there exists phenomena such as "robust overfitting" [1] and "catastrophic overfitting" [2]. The authors should clarify if they are proposing something new or just referring to either of them. If latter, I would suggest the authors to avoid the creation of new term and adopt the existing.
2. The motivation of the proposed method is weak. The paper could benefit from a more in-depth discussion of how updating critical layers, instead of all, mitigates adversarial overfitting.
3. Technically, the fine-tuning loss proposed in section 3.2 is similar to the loss function of TRADES.
4. 10-step PGD without restart is too weak so the conducted adversarial evaluation using it is unreliable. It is therefore unable for me to judge the soundness of the results e.g. in the Tab. 1.
5. It has been demonstrated that data augmentation methods such as IDBH [3] and generative models [4] are effective in mitigating overfitting in adversarial training. It could enhance the strength of the proposed method if it improves the performance when combined with them.

[1]  Rice, Leslie, Eric Wong, and Zico Kolter. "Overfitting in adversarially robust deep learning." International conference on machine learning. PMLR, 2020.

[2] Wong, Eric, Leslie Rice, and J. Zico Kolter. "Fast is better than free: Revisiting adversarial training." International Conference on Learning Representations. 2020.

[3] Li, Lin, and Michael W. Spratling. "Data augmentation alone can improve adversarial training." The Eleventh International Conference on Learning Representations. 2023

[4] Wang, Zekai, et al. "Better diffusion models further improve adversarial training." International Conference on Machine Learning. PMLR, 2023.

**Questions:**

Please refer to the above.

---

> ### Author Response · Authors · 2024-11-23
> **Response to Reviewer oD5s**
>
> ### **W1: Robust overfitting**
> Sorry for the confusion. We are discussing and tackling the same phenomena as mentioned in the two papers you provided. We have unified our wording to “robust overfitting”, or simply “overfitting”, in the revised paper.
>
> ### **W2: Motivation**
> We hypothesize that only specific critical layers in a network should be further robustified via adversarial training, and focusing on these layers, rather than the entire model or a larger subset, improves results and mitigates overfitting. With the proposed criticality metric in Equ. (6), we identify layers that are still learning non-robust features in the adversarial training process, and propose to sepcifically reduce their criticality with the rest of the model frozen. This layer-selective tuning enables the critical layers to learn robust features, while preventing the other already-robust layers to overfit on the adversarial training objective.
>
> Our hypothesis is supported by our results and reinforced by the ablation study, which shows that fine-tuning either too many or too few layers negatively impacts performance. Figure 3 demonstrates the importance of selecting the optimal number of critical layers for adversarial accuracy. Clean accuracy trends similarly, as the same number of critical layers is optimal for both adversarial and clean accuracy, and fine-tuning too many layers degrades performance in both cases. These results are shown in Appendix D.4, Figure 7.
>
> ### **W3: TRADES**
> CLAT differs significantly from the TRADES objective in both methodology and ultimate goal.
>
> CLAT focuses on fine-tuning specific layers identified as critical, while freezing the rest of the model. This selective optimization mitigates overfitting and enhances robustness without making global changes.
>
> In contrast, TRADES optimizes the entire network to reduce the difference between clean and adversarial outputs. This is a broader approach aimed at achieving robustness across the entire network, not just specific critical layers.
>
> #### **Mathematical Differences**
>
> The two objectives are fundamentally different:
>
> - In **CLAT**, the additional term $\lambda \mathcal{L}_C(f_S)$ targets the identified critical layers to reduce their sensitivity to perturbations. The objective $\mathcal{L}_C(f_S)$ is formulated in Equ. (8). This targets feature robustness specifically at the critical layers most prone to learning non-robust features, reducing their criticality without modifying the rest of the model.
>
> - In **TRADES**, the second term
>
>   $$\lambda \cdot \max_{x' \in \mathcal{B}(x)} \mathcal{L}(f_\theta(x'), f_\theta(x))$$
>
>   focuses on minimizing the output discrepancy between clean and adversarial examples. This involves finding adversarial examples through an inner maximization and ensuring that the network's response to these examples is similar to its response to clean data, thereby enhancing adversarial robustness.
>
> ### **W4: Stronger attacks**
> To verify the reliability of our evaluation, we provide additional results with increased steps of PGD, as well as additional attack methods, in Appendix D.7 (Table 17). CLAT shows consistent performance improvement over baseline under all attacking scenarios. Furthermore, we already show the performance of CLAT under varying epsilon values on different networks in Appendix B and include AutoAttack evaluations in the main paper, showing our performance improvement is universal.
>
> ### **W5: Data augmentation**
> While augmentation techniques have proven effective in mitigating adversarial training, we emphasize that this represents an orthogonal line of research, focusing on data, whereas our work is centered on model and objective optimization. Nonetheless, we provide results demonstrating the performance of CLAT after applying augmentation techniques to illustrate its versatility, as shown in Appendix D.8.2. Applying CLAT with the Augmented data further improves model robustness compared to baseline methods.

---

> > ### Comment · Reviewer_oD5s · 2024-11-25
> >
> > Dear Authors,
> >
> > Thank you for providing detailed responses and additional experimental results. I now recognize the empirical performance of the proposed method. It is commendable that the method has been demonstrated to complement data augmentation techniques in addressing robust overfitting. Accordingly, I have decided to raise the score to 6. However, a higher score was not assigned due to ongoing concerns about the technical novelty of the proposed method and the insufficient clarity of the motivation.

---

> > > ### Author Response · Authors · 2024-11-25
> > > **Thank you!**
> > >
> > > Thank you for your positive feedback and support.

---

### Official Review · Reviewer_agZN · 2024-11-01

**Soundness:** 2
**Presentation:** 3
**Contribution:** 2
**Rating:** 5
**Confidence:** 3

**Summary:**

This paper proposes a method to address adversarial overfitting issues by optimizing only the parameters in critical layers during adversarial training, instead of all parameters. The authors argue that this approach can effectively alleviate the problem. Specifically, they identify critical layers as those that exhibit a stronger tendency to learn non-robust features or demonstrate higher sensitivity to adversarial input perturbations compared to other layers in the model. The authors empirically demonstrate the effectiveness of their method through experiments conducted on CIFAR10 and CIFAR100 datasets.

**Strengths:**

The numerical experiments in the paper yield promising results, and the method's general applicability is validated across multiple models. The approach presented in this paper is intuitive, and the writing is clear and easy to understand.

**Weaknesses:**

The method proposed by the authors appears to be empirically feasible but lacks theoretical justification. I have certain reservations about the authors' approach, as it appears to be not much different from simply reducing the learning rate. Notably, the authors have fix the maximum number of epochs in their experimental setup, rather than training until convergence, which deviates from conventional practices. If the method merely prevents the model from learning to fit, in order to achieve the so-called mitigation of overfitting, I question the significance of this paper.

**Questions:**

I suggest the authors further investigate the performance of the model after increasing the number of epochs, as well as compare the performance of the model after reducing the learning rate. Additionally, I recommend including the performance of the model without adversarial training in the table, to facilitate comparison for the readers.

---

> ### Author Response · Authors · 2024-11-23
> **Response to Reviewer agZN**
>
> Thank you for raising these insightful points. We address your concerns in two parts:
> 1. **Extended Training Epochs and Convergence**: In response to your suggestion regarding convergence, we extended the training duration to 150 adversarial epochs, using a cosine learning rate scheduler that decays the learning rate to zero by epoch 150. As shown in Figure 5 (Appendix D.1), models trained with PGD-AT show a consistent decline in performance with more adversarial training epochs, highlighting the overfitting effect. In contrast, models trained with CLAT improves both performance and robustness across all epochs, demonstrating the mitigation of overfitting and allowing the model to fit more with more epochs being trained.
> 2. **Learning Rate Reduction Comparison**: We also conducted experiments with a reduced learning rate, reducing it by a factor of 10 at the 70th epoch, as presented in Appendix D.2 (Figure 6). This setup included comparisons between CLAT with the original learning rate, CLAT with the reduced learning rate, and PGD-AT with the reduced learning rate, all trained to 150 epochs with eventual decay of the learning rate to zero. Across both clean and adversarial accuracies, CLAT consistently outperformed the other configurations, reinforcing its effectiveness beyond merely reducing the learning rate.
>
> To add more discussion on the learning rate reduction, reducing learning rate may boost the performance in the specific epoch, but does not prevent the model from further overfitting. CLAT, on the other hand, enables continued performance improvement throughout training epochs regardless of the learning rate used.
>
> Lastly, the orange line in Fig. 5 indicates the performance of CLAT from scratch without any prior adversarial training. We also show the performance of CLAT on clean pre-trained models in Appendix Section C.1 (Table 11). These results show CLAT mitigates overfitting and improves both accuracy and robustness throughout the finetuning epochs, no matter of the starting point.

---

> > ### Author Response · Authors · 2024-11-30
> > **Thanks, further questions or clarifications**
> >
> > Thank you so much for your insightful feedback and we appreciate your time greatly! Please let us know if we can provide any further clarification or answer any remaining questions.

---

### Official Review · Reviewer_jmSg · 2024-11-02

**Soundness:** 3
**Presentation:** 3
**Contribution:** 2
**Rating:** 6
**Confidence:** 4

**Summary:**

The paper presents an adversarial-finetuning algorithm that selectively updates specific parameters in a deep network. The algorithm relies on iteratively finding `critically vulnerable' parameters and then leveraging variants of adversarial training to robustly train deep networks. The fraction of critical parameters is often less than 4% of the total network, leading to better regularization and reduced overfitting. The paper presents empirical results for a variety of adversarial training approaches, as well as several ablation studies.

**Strengths:**

1. The CLAT formulation expands upon existing intuition of constraining the local lipschitz constant of layers as an adversarial defense, by restricting the number of trainable layers. The idea of using smarter regularization to reduce overfitting is quite interesting, and proves to be useful in the settings that the paper describes.

2. The results and ablations are quite thorough, and support the claims in the paper. I especially appreciate the results in Fig. 2 for clearly showing the advantages and disadvantages of CLAT over just pure adversarial training.

**Weaknesses:**

Some possible improvements are listed below:

1. The criticality metric is essentially an estimate of the ratio of local lipschitz constants across layers. While the use of the metric as a way to rank layers by vulnerability is interesting, it would be better to use standard terminology here rather than coming up with new ones. I also suggest adding a discussion comparing other lipschitz regularization approaches used as an adversarial defense.

2. The paper mostly conducts experiments on CIFAR-10/100 which are generally standard in the field of robustness. However, an experiment on ImageNet or a dataset of similar size and complexity would be a valuable addition. One hypothesis to test is if a dataset of additional complexity also requires a similar number of CLAT parameters. This would also further strengthen the claims from Table 8 which suggests that criticality is an architecture-specific phenomenon.

3. Another ablation that I find would be useful is to track the change in critical layer distributions across training. Since the algorithm re-identifies critical layers every 10 epochs, does the distribution change significantly over training? Specifically, are critical layers more prominent in earlier or later layers as training progresses. This may help further reduce the hyperparameter exploration required for CLAT.

4. Finally, a comparison of epoch wall-time for PGD-AT versus PGD-AT + CLAT would be interesting.

Overall, the results are quite interesting and encouraging, but I would appreciate it if the authors addressed other lipschitz regularization approaches for adversarial defense in the draft.

Minor nit: The plots often have very thin lines close to each other. I suggest increasing the line thickness and fontsize in the figures to make them more clear.

**Questions:**

1. What was the reasoning behind approximating $\frac{\partial F(x')}{\partial x'}$ with a uniform vector in Eq. 5?

---

> ### Author Response · Authors · 2024-11-23
> **Response to Reviewer jmSg**
>
> Thank you for your insightful review!
>
> ### **W1: Local lipschitz constants**
> We thank the reviewer for pointing us to the line of Lipschitz regularization. We agree that the formulation we reached in Equation (5) is similar to the local Lipschitz constants, and would love to make the connection in our revised paper.
>
> Meanwhile, from the adversarial robustness perspective, there seems to be less exploration on Lipschitz regularization approaches. We take inspiration from the curvature formulation as in Equ. (3), which is a more accurate measurement to feature robustness and would lead to a more accurate criticality metrics. However, as we use the criticality metrics as both layer selection criteria and layer finetuning objective (see Equ. (8)), having a first-order gradient term in the metric will make the optimization significantly more costly. Here we empirically find that the approximation in Equ. (5) is good enough for improving clean accuracy and robustness of CNN models. Yet the full derivation may be useful for future work to apply CLAT-like methods on more complicated architectures that require a more precise criticality measurement.
>
> ### **W2: Larger dataset**
> We have provided results for Imagenette (Appendix D.6.1, Table 15) and Imagenet (Appendix D.6.2, Table 16) in our rebuttal. We used the same setting as in the CIFAR training, such as selecting 5% of the layers and using a random batch of 50 data points for criticality measurements. We note that our improvements are consistent as the CIFAR experiments, strengthening the observation that criticality seems to be an architecture specific phenomenon.
>
> ### **W3: Critical layer change***
> Table 13 (Appendix D.3) presents the computed critical indices for DN121, RN50, and RN18. We recompute the critical indices at epochs 70, 80, and 90 during the fine-tuning phase (30 epochs). Prior to this, we conduct adversarial training without freezing any layers. We observe that the distribution of critical layers changes over the course of training, although there is no clear trend indicating whether they are more prominent in earlier or later layers as training progresses. The results from this experiment, combined with the results from the ablation study where we do not change critical layers at all indicate that dynamic layer selection is crucial for CLAT.
>
> Additionally, Table 14 (Appendix D.5) bolsters our reasoning for utilizing dynamic selection. Across the board, CLAT conducted with dynamic layers outperforms CLAT with fixed layer selection throughout finetuning.
>
> ### **W4: Wall time**
> Expanding on results from Table 10, one PGD-AT epoch on DN121 takes 67 seconds on CIFAR and one epoch for CLAT finetuning takes the same amount of time. This is not including critical index recomputation time, which is minimal compared to the adversarial training time. Even if we take a batch size of 100 here (Critical indices can be computed accurately for batches as small as 10) for computing critical indices (and we do this thrice), it adds only 3.15s to the training time every 10 training epochs.
> Consider a typical PGD-AT+CLAT training process with 70 epochs of PGD-AT pre training followed by 30 epochs of CLAT finetuning, the total training time would take 10.5 extra seconds over the 1.86 hours standard adversarial training time. This timing is fairly consistent with most other networks as well.
>
> ### **Line and font in figures**
> Thanks for pointing this out, we will be sure to fix this in the final version.
>
> ### **Q1: Approxmation in Equ. (5)**
> As we use the criticality metrics as both layer selection criteria and layer finetuning objective (see Equ. (8)), having a first-order gradient term in the metric will make the optimization significantly more costly. Here we empirically find that the approximation in Equ. (5) is good enough for improving clean accuracy and robustness of CNN models.

---

> > ### Comment · Reviewer_jmSg · 2024-11-27
> > **Thanks for the response**
> >
> > I thank the authors for their detailed response, and appreciate the new results.  However, there has been significant work on leveraging lipschitz regularization for adversarial robustness. For example, [1, 2] study the use of lipschitz regularization as a defense. Most randomized smoothing methods for robustness certification also rely on controlling the lipschitzness of the network. I also agree with Reviewer s6zB that the math in sec 3.1 while useful for perhaps motivating the approach, does not really add much to the clarity of the paper. I am going to keep my score for now, but I do suggest that the authors avoid introducing new terminology unless absolutely necessary.
> >
> >
> > [1] Hein and Andriushchenko, Formal Guarantees on the Robustness of a Classifier against Adversarial Manipulation, NeurIPS 2017.
> > [2] Finlay et al., Lipschitz regularized Deep Neural Networks generalize and are adversarially robust. (2018)
> >  https://arxiv.org/abs/1808.09540

---

> > > ### Author Response · Authors · 2024-11-27
> > >
> > > Thanks for following up and for the encouraging and well-informed feedback. We value constructive criticism and your effort to understand and challenge our work—this kind of exchange is vital to the research community.
> > >
> > > We have revised Sec. 3.1 following yours and reviewer s6zB’s suggestion. We relegated the overly-complicated Hessian derivation to the Appendix (now Appendix E) as an alternative motivation to avoid potential reader confusion. In the main text, we acknowledge the relationship between our "feature weakness" formulation and the local Lipschitz constants of the feature and have explicitly highlighted this connection (starting at line 138).
> > >
> > > Interestingly, Local Lipschitz Constant was introduced in your referred paper as an approximation to the gradient/curvature regularization, which concurs with our original motivation in deriving the feature weakness. Meanwhile, we agree with you that we can directly use the existing concept of Lipschitz without going through the approximation again.

---

> > > > ### Author Response · Authors · 2024-11-30
> > > > **Thank you!**
> > > >
> > > > Thank you so much for your insightful feedback and we appreciate your time greatly! Please let us know if we can provide any further clarification or answer any remaining questions.

---

### Official Review · Reviewer_s6zB · 2024-11-03

**Soundness:** 2
**Presentation:** 2
**Contribution:** 2
**Rating:** 3
**Confidence:** 4

**Summary:**

This paper proposes a **Criticality-Leveraged Adversarial Training (CLAT)** method to enhance adversarial robustness by keeping certain parts of the neural network fixed and only fine-tuning the layers that are fragile in terms of robustness. Experiments are conducted across various datasets and neural network structures.

**Strengths:**

1. The explanation of the concept is clear.

2. The presentation in the experimental section is well-organized and easy to follow.

**Weaknesses:**

1. This method is cumbersome as it requires inputting both clean and adversarial images to compute the difference.

2. In Section 3.1, numerous mathematical tools are introduced but ultimately seem unnecessary, as the final objective function minimizes the initial starting point. This makes much of the content in Section 3.1 redundant.

3. Experiments are conducted only on the CIFAR dataset with CNN structures. The experimental design should be expanded to include both small- and large-scale datasets, as well as ViT models.

**Questions:**

1. Some tables could use bold font to highlight the best results.

2. To demonstrate the effectiveness of the proposed Layer Criticality Index, an ablation study on fine-tuning different critical layers with small or large $C_{f_i}$ values would be helpful.

---

> ### Author Response · Authors · 2024-11-23
> **Response to Reviewer s6zB**
>
> Thank you so much for your review, we really appreciate your attention to detail and have carefully addressed all of your concerns/ questions.
>
> ### **W1: Cumbersome**
> We respectfully disagree. CLAT training does not increase the computational complexity compared to the standard adversarial training methods, so the total training time does not differ. The only additional cost of CLAT involves identifying critical layers with the criticality indices. The computational overhead for our critical indices is negligible.
> * In our investigation in Sec 4.4, we present the average time required to compute critical layers across various batch sizes using randomized data on DenseNet121. It's worth noting that smaller networks exhibit even shorter computation times. Overall, we find that criticality indices can be computed with batch size as small as 10 (See Table 10). Conservatively, if we consider a batch size of 50, our critical layers are derived from a mere 0.0008% of the training data. Adopting criticality estimation every 10 epochs, as proposed in our method, introduces neglectable additional complexity to the training process.
> * Lastly, estimating the criticality for all layers involves two forward passes: one with a batch of clean input and one with corresponding PGD adversarial examples against the model output. Importantly, both clean inputs and adversarial examples are readily available during the adversarial training process. As we empirically demonstrate that the criticality estimation process is insensitive to the selection of training data, we can simply record the hidden features of the final batch of adversarial training data for the criticality computation in the new layer selection cycle.
>
> ### **W2: Math in Sec. 3.1**
> Intuitively, a layer learning non-robust features will be influenced more by adversarial examples. This impact can be measured by the difference between the output features when facing a clean input vs. facing an adversarial input. In Sec. 3.1,we further make a theoretical connection between the output feature difference and the local curvature around a layer’s output, showing that the output MSE under attack, as in Equ. (5), is an approximation to the local loss curvature, which represents the robustness as derived in [Robustness via curvature regularization, and vice versa]. This derivation ensures Equ. (5) is a valid estimation for local feature weakness/robustness, which builds the foundation for deriving the criticality metrics in Equ. (6).
>
> It should also be noted that Equ. (3), which comes with the first-order gradient term, is a more accurate measurement to feature robustness and would lead to a more accurate criticality metrics. However, as we use the criticality metrics as both layer selection criteria and layer finetuning objective (see Equ. (8)), having a first-order gradient term in the metric will make the optimization significantly more costly. Here we empirically find that the approximation in Equ. (5) is good enough for improving clean accuracy and robustness of CNN models. Yet the full derivation may be useful for future work to apply CLAT-like methods on more complicated architectures that require a more precise criticality measurement.
>
> ### **W3: Other datasets and models**
> As noted by reviewer jMsg, our experiments primarily focus on CIFAR-10/100, which is a standard benchmark in robustness research. However, we have extended our results to include performance on Imagenette and ImageNet, as presented in revised Appendix D.6
>
> ViT models, due to its larger size and diverse layer types, bring unique challenges to the feature-based layer selection process. This is out of the scope of this paper and we plan to address it in future work. Initial experiments on applying competing techniques such as RiFT and AutoLoRA on DeiT, ViT, and SwinB leads to more than a 3% reduction in adversarial accuracy (on PGD-10) compared to full adversarial training, showcasing the different characteristic of ViTs and CNNs.
>
> ### **Q1: Tables**
> Thank you for the feedback. We have revised accordingly
>
> ### **Q2: Layer selection ablation**
> We provide comprehensive ablations in the manuscript to demonstrate the effectiveness of our Layer Criticality Index. Notably, we compare the performance of selecting layers based on our critical indices versus random selection in Tab 12 (Appendix C.2), as well as dynamic versus static critical indices during CLAT in Tab 14 (Appendix D.5). These results clearly indicate the effectiveness of our layer selection approach
>
> In our experiments, we fine-tune layers with the largest Cfi values, as these are identified as the most critical. We do not perform ablations on fine-tuning layers with the smallest Cfi values, as this aligns with the premise of RiFT, which targets the most robust layers and finetune them with clean training objectives. Notably, our method consistently outperforms RiFT across all benchmarks, validating the superiority of our approach

---

> > ### Comment · Reviewer_s6zB · 2024-11-26
> >
> > Thank you so much for addressing my concerns. I have now changed my opinion regarding the method of using both clean input and adversarial input. How does the time complexity of this approach compare with LoRA and RIFT? I would like to see a detailed comparison.
> >
> > Additionally, I still find the mathematical explanation somewhat convoluted and indirect. It might confuse readers by introducing unnecessary mathematical concepts. For example, the final formula (7) could be derived directly from equation (2) without the intermediate definitions, which seem overly complex only to be simplified later. A more direct approach would make it clearer and easier to follow.
> >
> > Lastly, I don’t quite understand why you did not perform ablations on fine-tuning layers with the smallest $C_{f_i}$ values. According to your Definition 3.1, a larger $C_{f_i}$ indicates that a layer is more critical, as it increases the weakness of the features after it. If this index is indeed meaningful, fine-tuning layers with smaller $C_{f_i}$ values should yield worse results compared to tuning critical layers, but still better than the vanilla model. I don’t think you can justify omitting this ablation study, even if you have already surpassed RiFT. Including this study is crucial to fully demonstrate the effectiveness of your criticality index.
> >
> > I will keep my scores unchanged for now.

---

> > > ### Author Response · Authors · 2024-11-27
> > >
> > > Thank you for your encouraging feedback. We truly appreciate the detailed follow-ups, constructive criticism, and your dedication to understanding and challenging our work. This kind of engagement embodies the spirit of this discussion period and, more broadly, the research community.
> > >
> > > 1. We have included a more detailed time comparison for RiFT, CLAT and AutoLoRA in Appendix D9, with DN-121 on CIFAR-10. In summary, CLAT and AutoLoRA are marginally slower than PGD-AT (69 vs 67 second/epoch), potentially due to the overhead in data movement. RiFT is much faster at 56 second/epoch, benefiting from using clean training objectives in finetuning. However, the use of clean objectives also hinders RiFt’s ability to further improve robustness in the finetuning process. In the reported experiments, RiFT used 110 epochs of adversarial pretraining to get adequate robustness, while CLAT uses only 70\. Consequently, the total training time for RiFT is 132 minutes (110 epochs of adversarial training and 10 epochs of fine-tuning), compared to 112 minutes for CLAT (70 epochs of adversarial training and 30 epochs of fine-tuning), yet still achieving worse accuracy and robustness.
> > > 2. While we agree with the reviewer that the mathematical details regarding the formulation and simplification of curvature may be over complicated and not directly enhance clarity for all readers, we believe they still serve as valuable motivation (as noted by reviewer jmSg). To strike a balance, we have moved these details to the appendix (now Appendix E), allowing interested readers to access them without detracting from the main narrative. Additionally, we now comment on the connection between our weakness formulation and local Lipschitz constants of the feature function (starting at line 138\) to make the derivation more direct and focused.
> > >    We also want to emphasize two key components of our method: “identifying critical layers with criticality” and “optimizing critical layers in finetuning”. The definition of Weakness and Criticality plays a crucial role in how we select critical layers, so we have retained its mention rather than removing everything between the previous Equ. (2) and Equ. (7) (Equ (4) now in the revised paper) as you suggested. We believe this strikes the right balance between the integrity and conciseness of our method.
> > > 3. We acknowledge that RiFT is a distinct approach, and the significance of our Cfi is better clarified through this ablation study of finetuning the "least critical" using CLAT objective, which we have now included in Appendix D.10. Thank you for pointing this out. As expected, fine-tuning layers with smaller Cfi values yields worse results compared to tuning the critical layers with largest criticality but still performs better than the vanilla model.

---

> > > > ### Author Response · Authors · 2024-11-30
> > > > **Thanks, questions or further clarifications**
> > > >
> > > > Thank you again for your insightful feedback and appreciate your time greatly. Please let us know if we can provide any further clarification or answer any remaining questions.

---

### Official Review · Reviewer_sWy4 · 2024-11-03

**Soundness:** 2
**Presentation:** 3
**Contribution:** 2
**Rating:** 6
**Confidence:** 4

**Summary:**

The paper tackles the important issue of adversarial vulnerability of neural networks. Instead of updating all parameters during adversarial training, CLAT focuses on layers that predominantly learn non-robust features, identified by a "criticality index." This selective focus is designed to improve adversarial robustness, can be used with on top of standard adversarial training approaches. The paper offers detailed results on multiple architectures and also evaluated under both white-box and black-box adversarial attacks

**Strengths:**

- CLAT presents an alternative to traditional adversarial training, which requires updating the entire network, and can be used on top of any adversarial training method
- The analysis is detailed and covers many network architectures
- The method is evaluated on various attacks and also includes the strong attacks such as Auto Attack
- The ablation study and the overhead analysis is good and useful

**Weaknesses:**

- I am unclear about the problem statement and objective. Because, in the Adversarial domain, there are two main adversarial training challenges: (1) the trade-off between clean and adversarial accuracy and (2) adversarial overfitting, and there are methods that cater to either or both, but the evaluation criteria are different. For ex, for trade-off we need to check how clean acc and adv acc are impacted, and for the latter, we need to report both the peak and final accuracy scores, highlighting the impact of overfitting.
- The concept of robust and non-robust features is important in the paper, and needs more context and explanation. What is the criteria to differentiate, and further an empirical example or analysis will provide more clarity.
Additionally, the theoretical explanation behind the criticality index would benefit from more intuition. Is the motivation based on the paper [Robustness via curvature regularization, and vice versa], that small curvature correlated to higher robustness? How does this relate to criticality or non-robust features? The main focus of the paper about critical layers and its criterion is unclear to me.
- The paper uses multiple terms—such as "weakness," "robust-critical layers," "sensitive layers," and "non-robust features". This variation makes it difficult to understand the precise characteristics, need consistent definitions. Also are we talking about NR layers (weights) or features, there's multiple usages of these terms.
- Since CLAT fine-tunes layers that supposedly learn non-robust features (which aid clean accuracy), one would expect that this selective fine-tuning might reduce clean accuracy, however the clean accuracy improves as well ? I am curious to understand more on this.
- Line 430: Additional intuition or analysis is needed to support this claim. What architectural properties might influence these findings? Additionally, is there a discernible pattern to the critical layers? Identifying such patterns could provide valuable insights into what different layers learn and how this understanding could be leveraged to improve robustness further.

**Questions:**

- Could you elaborate on different hyperparameters and how they were chosen? Ex. the 5% threshold for selecting critical layers. Is there any consideration for adjusting the percentage based on the depth or structure of the network?
- Robust/Non-Robust - is it only layer wise, or also makes sense to see sub-networks or neuron-wise? Introduction could use more context
- Results - (related to point 1 in weaknesses), if the the paper claims to address overfitting in adversarial training, it would be essential to compare CLAT with explicit overfitting reduction techniques like AWP (Adversarial Weight Perturbation) or SWA (Stochastic Weight Averaging). Including metrics such as best vs. final accuracy scores for these methods would help demonstrate whether CLAT effectively mitigates overfitting.
- Computational overhead on bigger and more complex datasets? (other than CIFAR)
-  Line 461 - Did not understand the hypothesis for the behavior of Figure 3. Please elaborate with supporting evidence. Further, how is the behavior on clean accuracy? In general, the ablation study is interesting, but the claims and intuitions presented would benefit from more supporting evidence to strengthen their validity.

---

> ### Author Response · Authors · 2024-11-22
> **Response to Reviewer sWy4 part 1**
>
> ### **Problem statement and objective**
> Overfitting is the primary challenge we address, but we caveat that this encapsulates the degradation of both clean and adversarial accuracy. Overfitting occurs when the model becomes excessively tailored to adversarial examples, reducing its ability to generalize and impacting both clean accuracy and robustness over time.To accentuate this, we extend the learning curve experiments in Figure 2 to 150 adversarial training epochs, using a cosine learning rate scheduler that decays the learning rate to 0 by epoch 150. As shown in **Figure 5 in the updated Appendix D.1**, models trained with PGD-AT exhibit a consistent decline in performance with additional adversarial training epochs, clearly illustrating the effects of overfitting. In contrast, models trained with CLAT maintain consistent performance and robustness improvements across all epochs. We note that CLAT models also achieve higher peak and final accuracies as a result of being significantly less prone to overfitting.
>
> ### **Criticality and non-robust features**
> The concept of robust and non-robust features is indeed crucial, and we appreciate the opportunity to provide more context. Non-robust features are those that significantly impact adversarial robustness, as they are highly sensitive to adversarial perturbations. Intuitively, a layer learning non-robust features will be more affected by adversarial examples. This sensitivity can be measured by the difference between the layer's output features when faced with a clean input versus an adversarial input.
>
> In Section 3.1, we further establish a theoretical connection between this output feature difference and the local curvature around a layer's output. Specifically, we show that the output MSE under attack, as defined in Equation (5), serves as an approximation to the local loss curvature, which correlates with robustness as derived in [Robustness via curvature regularization, and vice versa].
>
> To isolate the contribution of each layer to the learning of non-robust features, we define a layer criticality measure in Equation (6), which represents the ratio of the feature MSE computed before the layer to the feature MSE computed after the layer. Layers with higher criticality contribute more significantly to the learning of non-robust features, aligning with our definition of "critical layers" in Definition 3.1. Empirical results in Table 7 demonstrate that selecting these critical layers is crucial for the robustness and accuracy improvements achieved by CLAT.
>
> ### **Weakness, critical, and sensitive**
> We appreciate the reviewer's feedback and apologize for the confusion caused by inconsistent terminology. To clarify:
> * "Weakness" refers to a property of the feature, as defined in Equation (5), whereas "criticality" refers to a property of the layer, as defined in Equation (6). We use the term "weakness" as a concise noun to describe how non-robust a feature is. Since "non-robust" is an adjective and "non-robustness" feels too cumbersome, "weakness" serves as a fitting alternative as it is the opposite of "robustness."
> * We acknowledge that the term "sensitive" was unnecessary and led to ambiguity. As a result, we revised Section 3.1 to eliminate the use of "sensitive" for greater clarity.
> * We now use consistent terminology throughout the paper: in adjective form, we refer to a "non-robust feature" and a "critical layer," while in noun form, we use "feature weakness" and "layer criticality." These changes are clearly reflected in the revised Section 3.1 and aim to eliminate any ambiguity regarding the terms.
>
> Additionally, we have ensured consistent usage of the terms “layer” and “feature” to avoid confusion between non-robust features and critical layers. The revised manuscript now clearly differentiates between features and layers and uses these terms consistently.

---

> ### Author Response · Authors · 2024-11-22
> **Response to Reviewer sWy4 part 2**
>
> ### **Clean accuracy improvement**
> To achieve an adversarially robust model, we want all layers to learn robust features. Note that well-learnt robust features should also be useful, which leads to both robust and accurate classification. We believe an ideal conversion from learning non-robust features to robust features should not lead to clean accuracy drop. A model can be both robust and accurate, much like the human brain.
>
> We hypothesize that overfitting is the true cause of loss of clean accuracy in standard adversarial training. In other words, we believe that overfitting to adversarial examples can degrade the model’s performance on clean data. CLAT is designed to counter this issue by selectively fine-tuning only the critical layers that need robustness improvements, while keeping already-robust layers fixed. We propose that this approach prevents overfitting to adversarial data and ensures that the model retains its ability to perform well on clean inputs.
>
> Our empirical results suggest that this selective fine-tuning approach effectively mitigates overfitting during adversarial training, allowing the critical layers to transition from learning non-robust features to learning robust ones without any loss of clean accuracy—and even leading to improvements. This outcome supports our hypothesis that improving robustness can indeed go hand-in-hand with improving clean accuracy, distinguishing CLAT from standard adversarial training methods that typically face a tradeoff between accuracy and robustness.
>
> ### **Architectural property**
> We appreciate this insightful question. Line 430 is what we observe empirically, and we acknowledge that the exact architectural properties influencing these findings remain unclear. Identifying patterns in critical layers and understanding the architectural properties that contribute to these results is indeed an important area for future research. We explicitly highlight this as an open question (in Conclusions) and a promising direction for future work.
>
> ### **Hyperparameters**
> The three primary hyperparameters in our approach are:
> 1. **Percentage of Critical Layers**: For each network, we conducted experiments where the algorithm was applied to 1, 2, ..., up to the total number of layers that could be selected as critical (entire fine-tuning). As stated in Lines 460–464, “Allowing more layers to be fine-tuned enhances the model flexibility, which initially improves CLAT performance. However, fine-tuning more layers diminishes CLAT's effectiveness. This is likely due to the diversion of attention towards fine-tuning less-critical layers, detracting from more critical ones.” Based on this, 5% was selected as a general threshold that balances performance across different architectures. We show the robustness-layer number curve in Fig 3 and the accuracy-layer number curve in Fig. 7 (revised Appendix D.4), which reflects our observations.
> 2. **Number of Epochs for Fine-Tuning**: Based on Figure 2, fine-tuning for 30 epochs appears to be optimal. This observation is consistent with results obtained when fine-tuning is extended to 150 epochs, where the learning rate decays to 0, across all tested networks.
> 3. **Frequency of Computing Critical Indices**: The frequency of recomputing critical indices (every 𝑋 epochs) was determined by testing the variance in critical layers after every epoch of training. We found that critical layers typically take approximately 10 epochs to become less critical through CLAT, making 10 epochs an effective interval for recomputation.
>
> ### **Subnetworks**
> The observation regarding robustness/non-robustness at sub-network or neuron-wise levels is an interesting direction. However, we focus on layerwise adjustments because they represent the outermost lever in the optimization space, offering significant performance improvements with manageable complexity.
>
> We initially explored alternative granularities, such as blockwise and channelwise adjustments. Blockwise adjustments did not provide any gains, while channelwise adjustments introduced substantial computational overhead due to the significantly higher number of structures involved. Given these findings and the strong performance achieved with layerwise adjustments, we chose not to pursue finer-grained methods further.

---

> ### Author Response · Authors · 2024-11-23
> **Response to Reviewer sWy4 part 3**
>
> ### **Overfitting reduction techniques**
> CLAT proposes a way to mitigate overfitting by selecting and updating only the critical layers. We believe CLAT is orthogonal to other explicit overfitting reduction techniques that are applied in the full model optimization process. Results for combining CLAT with other explicit overfitting reduction techniques are shown in Appendix D.8.1, Table 18, where final performance is reported. CLAT works together with the mentioned methods and improves performance universally.
>
> ### **Computational overhead**
> We want to clarify that our method does not actually have any computational overhead compared to standard PGD-AT except for the times required to compute critical indices (which is trivial as shown in Table 10). No matter the dataset or model used, criticality can be stably computed with a single training batch. In our ImageNet experiments in Appendix D.6.2 we find a batch size of 50 is adequate for criticality measurements. This cost is negligible compared to the adversarial training cost of the full training set.
>
> ### **Behavior of Figure 3**
> This is addressed above in the discussion on hyperparameter selection. To reiterate, Figure 3 demonstrates the importance of selecting the optimal number of critical layers for adversarial accuracy. Clean accuracy trends similarly, as the same number of critical layers is optimal for both adversarial and clean accuracy, and fine-tuning too many layers degrades performance in both cases. These results are shown in Appendix D.4, Figure 7.
>
> We hypothesize that only specific critical layers in a network should be further robustified via adversarial training, and focusing on these layers, rather than the entire model or a larger subset, improves results and mitigates overfitting. This hypothesis is supported by our results and reinforced by the ablation study, which shows that fine-tuning either too many or too few layers negatively impacts performance.

---

> > ### Comment · Reviewer_sWy4 · 2024-11-25
> >
> > Thank you for your detailed responses and the new results.
> >
> > - I still have some confusion on the terminology. When you use "overfitting", it is more on terms on generalization performance to clean and adversarial accuracy? Because "Robust Overfitting" is different phenomenon, where the test adversarial accuracy starts decreasing after a point during adversarial training, which is not a behaviour usually seen in standard training (on clean images). Check [1] and [2], and hence these papers provide best test accuracy and final test accuracy, to show the effect of robust overfitting. [3] works on both, generalization trade-off (between clean and adversarial) and also robust overfitting.
> > From Figure 5, does it imply that CLAT also mitigates robust overfitting? and the original goal was generalization?
> > Hence, I again wanted to clarify on this. Thanks.
> >
> > - Hyperparamters - Has the 5% threshold worked as the best results parameter for all networks used in the paper? Can there be a criterion to determine this for a architecture/dataset during warmup maybe?
> >
> > [1] Overfitting in adversarially robust deep learning
> > [2] Robust Overfitting may be mitigated by properly learned smoothening
> > [3] Conserve-Update-Revise to Cure Generalization and Robustness Trade-off in Adversarial Training

---

> > > ### Author Response · Authors · 2024-11-25
> > >
> > > Thank you for your timely reply\! Here we would like to provide further clarification for your questions
> > >
> > > ### **Robust Overfitting**
> > >
> > > This work is motivated by the phenomenon of robust overfitting. As observed in your citation \[1\], robust overfitting happens in the later stage of adversarial training, which is where CLAT takes effect. We observe that standard adversarial training will lead to a small portion of critical layers to still learn non-robust features, while the rest of the model, which is already robust, keep overfitting. To this end, CLAT selects and finetunes only the critical layers while freezing the rest already-robust layers, therefore preventing overfitting while improving the model robustness.
> > >
> > > As discussed previously for your comments on clean accuracy improvement, we believe the loss of clean accuracy is also contributed by the robust overfitting, where the model fits towards the training adversarial example and deviates from clean validation distribution. To this end, CLAT also helps generalization as we tackle the robust overfitting phenomenon.
> > >
> > > One additional note on the reporting of best and final accuracy. We are well aware of the effectiveness of early stopping in standard adversarial training, so we report the early stop performance at epoch 100 in all main tables in the paper, where standard adversarial training achieves the “best” performance, for a fair comparison. As shown in the full learning curve in Fig. 6, the accuracy and robustness of standard PGD-AT (purple line) become flat around epoch 80-100, and start dropping afterwards to a much lower final accuracy. Whereas CLAT achieves higher final accuracy with more finetuning epochs as we mitigate robust overfitting.
> > >
> > > ### **Hyperparameters**
> > >
> > > We empirically find that the 5% threshold works well across different models and datasets. From the ablation in Fig 3, we note that though the number of layers chosen would affect the learnt robustness, CLAT can achieve robustness gain over early-stopping baseline (no finetuning) with up to 10% of the layers chosen, showing that CLAT is stable under small changes of layer selection numbers.
> > >
> > > Also, CLAT employs a multi-round critical layer identification and finetuning process. As we observe that selecting too many layers, which leads to overfit, is often more harmful than selecting too few layers, which leads to underfit/slower convergence, we can reduce the number of layers selected in a new round if we see adversarial robustness or clean accuracy start dropping in previous epochs. This would help recovering the model from overfitting and enable further improvements.

---

> > > > ### Author Response · Authors · 2024-11-30
> > > > **Thanks, further questions or clarifications**
> > > >
> > > > Thank you so much for your insightful feedback and we appreciate your time greatly! Please let us know if we can provide any further clarification or answer any remaining questions.

---

> > > > > ### Comment · Reviewer_sWy4 · 2024-12-02
> > > > >
> > > > > I have increased my score to 6. I would recommend re-organizing the story and motivation, as it is still bit confusing to me as in the paper, it seems the motivation is to solve the generalization trade-off, however during the discussion you mention it was robust overfitting. More clarity on the motivation, empirical analysis to substantiate and clear ultimate goal would be beneficial.

---

> > > > > > ### Author Response · Authors · 2024-12-02
> > > > > >
> > > > > > Thank you for your positive feedback and support! We greatly appreciate your effort in helping us improving the quality and clarity of this paper in the discussion period.

---

### Author Response · Authors · 2024-11-22
**Summary of paper revision**

We thank the constructive feedback from all reviewers. The submitted paper has been revised to incorporate your suggestions. In the main paper, we fixed wording issues that lead to the confusion of reviewers sWy4 and oD5s. We also updated the formatting of some tables following the suggestions of the reviewer s6zB. We added a new **Appendix D** to the paper, which consolidated all additional experimental results suggested by the reviewers for easier reference. We will integrate these results and our discussions in the rebuttal text into the finalized version.

---

### Author Response · Authors · 2024-12-02

We thank the help and support from all reviewers in improving our paper in this discussion period. We would like to kindly remind the reviewer that the discussion period will conclude soon. We hope our responses and revisions help addressing your concerns. If further clarification is needed, please do not hesitate to ask.

---

### Author Response · Authors · 2024-12-04
**Thanks, final note on revisions and clarifications**

**We sincerely appreciate your thoughtful feedback and the opportunity to refine our work during the review process.** We believe that CLAT marks a significant advancement in adversarial training and has the potential to benefit the research community substantially. The primary motivation of this work is to mitigate robust overfitting during adversarial training, thereby improving both clean and adversarial accuracy. Moreover, CLAT integrates seamlessly with existing adversarial training methodologies to deliver state-of-the-art performance. We are pleased that all reviewers acknowledged the strong empirical performance of CLAT and appreciated the thoroughness and organization of our experiments and ablation studies.

In response to your feedback, we have implemented several key revisions. We ensured consistent terminology throughout the paper and added time comparisons with RiFT and AutoLoRA. Additionally, we conducted an ablation study to further validate the effectiveness of our index metric. To improve clarity, mathematical derivations were reorganized, with some details moved to the appendix. Section 3.1 was updated to incorporate your input, and we clarified the relationship between our "feature weakness" formulation and local Lipschitz constants. Concerns regarding extended training and reduced learning rates were also addressed through comprehensive empirical analysis.

We are confident that these revisions address all concerns and **would be deeply grateful if you find them satisfactory and consider awarding higher scores.** Thank you again for your valuable insights and engagement.

---

### Meta-Review · Area_Chair_Ldd3 · 2024-12-20

**Metareview:**

This paper proposes a technique to improve the robust overfitting using a technique called CLAT. Instead of updating all parameters during adversarial training, CLAT focuses on layers that predominantly learn non-robust features, identified by a "criticality index.". The fraction of critical parameters is often less than 4% of the total network, leading to better regularization and reduced overfitting

During the review period, there are several concerns raised about lack of thorough experimentation including datasets and models. Some reviewers also point to the lack of novelty of the approach, and that the paper is unclear to follow.

During the rebuttal, authors have improved the writing taking into account the reviewer's feedback. However, I think the authors still need to improve the paper and make it easy to understand. While the reviewers added experiments on imagenet during the rebuttal, I still feel a solid comprehensive comparisin and ablation studies are still missing, The fact that the method doesn't work with ViTs yet is another issue.

Overall, I feel the authors need to improve the experimentation, positioning and paper writing. So, I feel the paper is not ready for publication yet and I encourage the authors to take into account the invaluable suggestions given by the reviewers to improve the paper.

**Additional Comments On Reviewer Discussion:**

The reviewers raised a lot of concerns about the lack of clarity in the paper, lack of a clear problem statement and objective and missing experiments. The authors addressed several of these concerns in the rebuttal, but I feel they are not fully addressed. I feel the paper can get into a better shape with more experimentation and improving the clarity in writing. So, I decide to vote for rejecting the paper since it is not in. a fully publishable state for a conference like ICLR.

---

### Decision · Program_Chairs · 2025-01-22

Reject